# Microglia permit climbing fiber elimination by promoting GABAergic inhibition in the developing cerebellum

Hisako Nakayama[1,2], Manabu Abe[3,4], Chie Morimoto[1,5], Tadatsune Iida[6], Shigeo Okabe [6], Kenji Sakimura[3,4] & Kouichi Hashimoto [1]

Circuit refinement during postnatal development is finely regulated by neuron–neuron interactions. Recent studies suggest participation of microglia in this process but it is unclear how microglia cooperatively act with neuronal mechanisms. To examine roles of microglia, we ablate microglia by microglia-selective deletion of colony-stimulating factor 1 receptor (*Csf1r*) by crossing floxed-*Csf1r* and *Iba1-iCre* mice (*Csf1r*-cKO). In *Csf1r*-cKO mice, refinement of climbing fiber (CF) to Purkinje cell (PC) innervation after postnatal day 10 (P10)–P12 is severely impaired. However, there is no clear morphological evidence suggesting massive engulfment of CFs by microglia. In *Csf1r*-cKO mice, inhibitory synaptic transmission is impaired and CF elimination is restored by diazepam, which suggests that impairment of CF elimination is caused by a defect of GABAergic inhibition on PCs, a prerequisite for CF elimination. These results indicate that microglia primarily promote GABAergic inhibition and secondarily facilitate the mechanism for CF elimination inherent in PCs.

[1] Department of Neurophysiology, Graduate School of Biomedical and Health Sciences, Hiroshima University, 1-2-3 Kasumi, Minami-ku, Hiroshima 734-8551, Japan. [2] Department of Physiology, School of Medicine, Tokyo Women's Medical University, 8-1 Kawada-cho, Sinjuku-ku, Tokyo 162-8666, Japan. [3] Department of Cellular Neurobiology, Brain Research Institute, Niigata University, 1-757 Asahimachi-dori, Chuo-ku, Niigata 951-8585, Japan. [4] Department of Animal Model Development, Brain Research Institute, Niigata University, 1-757 Asahimachi-dori, Chuo-ku, Niigata 951-8585, Japan. [5] Department of Psychosocial Rehabilitation Graduate School of Biomedical & Health Sciences, Hiroshima University, 1-2-3 Kasumi, Minami-ku, Hiroshima 734-8551, Japan. [6] Department of Cellular Neurobiology, Graduate School of Medicine, University of Tokyo, 7-3-1 Hongo, Bunkyo-ku, Tokyo 113-0033, Japan. These authors contributed equally: Hisako Nakayama, Manabu Abe. Correspondence and requests for materials should be addressed to K.H. (email: hashik@hiroshima-u.ac.jp)

Most microglia derive from primitive myeloid progenitors that originate in the extraembryonic yolk sac and penetrate into the brain from embryonic day 8.5 in mice[1]. Microglia dynamically change their distribution and morphology during development[2–8]. Ramified microglia evenly distribute across the entire adult brain irrespective of whether it is gray or white matter (WM), but distribution of immature microglia with ameboid morphology is biased to the medulla and ventricular/subventricular zones around the time of birth[2–5,7–9]. Because microglia in the vicinity of targets act by direct contact,

release of bioactive molecules or phagocytosis[10,11], their influential range is relatively restricted. Therefore, the functional roles of microglia likely change with the progress of their translocation in the developing brain[8, 12], and it is important to analyze the functional roles of microglia in association with changes in their spatial distribution.

Microglia play indispensable roles in shaping neuronal circuits in various ways in the developing central nervous system. They regulate the number of neurons by promoting apoptotic cell death[13] and removal of neural precursors by phagocytosis[5]. In addition, microglia remodel synaptic connections. Neuronal circuits are initially redundant at birth but are gradually refined during postnatal development. Many studies have proposed that these developmental processes are regulated by an interplay between neurons[14,15]. However, recent studies have disclosed that microglia promote spine and excitatory synapse formation[16–18] and elimination of redundant excitatory synapses[18–23] during postnatal circuit refinement. Some synaptic connections are thought to be removed by engulfment by microglia[19–23], but it is largely unclear how microglia participate in a coordinated manner in postnatal circuit refinement regulated by neuronal mechanisms.

The cerebellar cortex is an appropriate model system to study the mechanisms for constructing functional neuronal connections. The majority of cerebellar Purkinje cells (PCs) receive inputs from only one climbing fiber (CF) in the adult but are innervated by multiple CFs at birth. One CF is selectively strengthened on the PC soma relative to the others by postnatal day 7 (P7)[15]. The single winner CF then undergoes translocation to the PC's dendrites[24] and extensive parallel fiber (PF) synapse formation proceeds[25]. In parallel, redundant CFs are eliminated[15,26] and PC somata are instead innervated by inhibitory basket cells[27]. The construction of cerebellar circuits is thought to be finely regulated by neuron–neuron interactions[28–30]. Although factors necessary for such neuron–neuron interactions have been extensively studied[15,28–30], the roles of non-neuronal cells, such as glial cells, are largely unclear.

In the present study, we mainly focused on the postnatal development of mouse CF–PC synapses to elucidate the roles of microglia in neuronal circuit development. For this purpose, microglia were ablated from the developing cerebellum by cerebellar injection of clodronate liposomes or by microglia-selective deletion of the colony-stimulating factor 1 receptor (Csf1r) gene (Csf1r-cKO). We found that CF synapse elimination is severely impaired in these mice. Importantly, inhibitory synaptic transmission was attenuated in Csf1r-cKO mice and CF elimination was restored by administration of a GABA$_A$ receptor sensitizer, diazepam. These results suggest that microglia primarily enhance GABAergic synaptic transmission to PCs during postnatal development, thereby indirectly promoting the process of CF synapse elimination that is controlled by the interplay between inhibitory neurons and PCs.

## Results

### Biased distribution of microglia beneath the Purkinje cell layer (PCL) at P8–P9.

We examined the developmental changes in microglial distribution in the cerebellum. At P5, the density of microglia was much higher in the WM than in the gray matter (internal granule cell layer: IGL, Purkinje cell layer: PCL, and molecular layer: ML) (Fig. 1a, b), consistent with a previous report[7]. The majority of the Iba1-positive microglia in the WM at P5 were ameboid, with large somata that were devoid of long processes (Supplementary Fig. 1). After P8–P9, the localization of microglia to the WM gradually decreased in parallel with the growth of the cerebellar cortex[31] (Fig. 1a–d), to finally become almost evenly distributed in the cerebellum at P60 (Fig. 1a, b). The morphology of the Iba1-positive microglia in the gray matter also changed after P8–P9. Their processes became thinner and more branched, and their somata became smaller as development proceeded (Supplementary Fig. 1). As previously suggested in quail and rat cerebellum[4,7], invasion of microglia into the developing cerebellar gray matter mainly occurred from the WM, because the density of microglia was higher on the WM side of the IGL and the external granule cell layer (EGL) was almost devoid of microglial somata (Fig. 1a, b).

Notably, we found that the density of microglial somata was significantly higher around the area from the PCL to the IGL that peaked beneath the PCL at P8–P9 (Fig. 1e, left). This biased distribution was not observed at P13–P14 (Fig. 1e, right). We confirmed this biased distribution using higher-volume specimens (Fig. 1f). Wild-type mouse brains were optically cleared by SeeDB2[32], and the distribution of microglia was examined in a volume of $200 \times 500 \times 500 \ \mu m^3$ for each animal ($n = 3$ mice). At P8–P9, the density of microglial somata had a large peak beneath the PCL and a smaller one at the edge between the ML and the EGL (Fig. 1g). Taken together, this transient and biased distribution of microglia raises the possibility that they may affect postnatal development of neuronal circuits in the cerebellar cortex at P8–P9.

### Pharmacological ablation of microglia impairs CF elimination.

The number of CFs innervating a single PC gradually increases during the first postnatal week, but starts to decrease after P7 in the mouse cerebellum[33]. The transient microglial accumulation we observed beneath the PCL is almost coincident with the start of CF elimination, which suggests a contribution by microglia to CF elimination. To test this possibility, we ablated microglia by infusion of liposomal clodronate into cerebellar lobules VI/VII or VIII at P6 or P7, around the start of CF elimination in wild-type mice. Liposomal clodronate is engulfed by phagocytic cells and induces cell death[14]. Immunohistochemical staining revealed that microglia locally disappeared from the cerebellar cortex around the injection site by P8 (Fig. 2a–c). Severe morphological alterations of calbindin-labeled PCs were not apparent (Fig. 2a, b). Importantly, injection of empty liposomes did not deplete

**Fig. 1** Postnatal changes in microglial distribution in the cerebellum. **a** Immunofluorescent labeling for Iba1 (green) and calbindin staining (red) in lobules IV–V in the vermis of mouse cerebellum at P5, P8, P14, P21, and P60. *EGL* external granule cell layer, *ML* molecular layer, *PCL* Purkinje cell layer, *IGL* internal granule cell layer, *WM* white matter. Scale bars, 50 μm. **b** Frequency distribution histograms for the relative depth of microglia at P5 ($n = 5$ mice), P8–P9 ($n = 5$), P13–P14 ($n = 4$), P20–P21 ($n = 4$), and P60 ($n = 6$). The depth of microglia was normalized to the total length of the gray matter and the white matter where each microglial cell was sampled. **c** Postnatal changes in densities of microglia in the WM (blue) and the gray matter (orange). Numbers of mice at individual postnatal days are same as in (**b**). **d** Postnatal changes in the total length of the ML, PCL, IGL, and WM. **e** Frequency distribution histograms of the relative depth of microglia in the gray matter except the EGL (the ML, the PCL, and the IGL) at P8–P9 (left, $n = 5$ mice) and P13–P14 (right, $n = 4$). Note that microglia tend to aggregate beneath the part of the PCL (the area between orange broken lines) and the edge beside the WM at P8–P9; *$p < 0.05$, **$p < 0.01$. **f** Volume rendered image ($508 \times 508 \ \mu m$, 450 μm depth) of the mouse cerebellum at P8 with immunofluorescent labeling of Iba1 (green) and calbindin (red). **g** Density of microglia across the layers ($n = 3$ mice). Data are presented as mean ± SEM

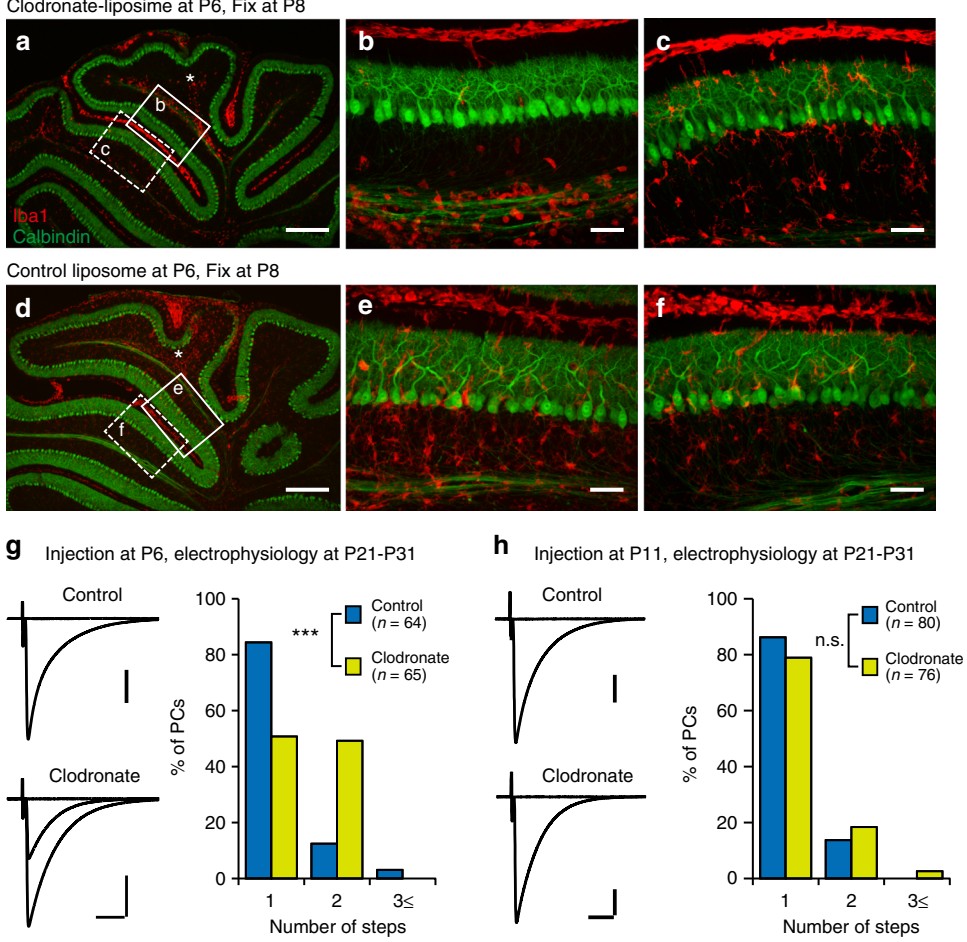

**Fig. 2** Effects of pharmacological deletion of microglia on CF synapse elimination. **a–f** Double immunolabeling for Iba1 (red) and calbindin (green) in liposomal clodronate (**a–c**) or control liposome (**d–f**) injected mice at P8. Asterisks in **a** and **d** indicate the injection sites of the liposomes in lobules VI–VII in the vermis. Regions surrounded by solid and dotted lines are magnified in **b** and **e**, and **c** and **f**, respectively. Iba1-labeled microglial cell numbers are significantly reduced in the clodronate-injected lobules VI–VII (**b**), but not in the neighboring lobules IV–V (**c**). **g, h** (left) Representative traces of CF-EPSCs recorded in mice treated with control (upper) or clodronate (lower) liposomes at P6 (**g**) or P11 (**h**). Electrophysiological recordings were conducted from PCs in the liposome-injected lobules at P21–P31. The holding potential (Vh) is −10 mV. Scale bars, 1 nA (**g**), 0.5 nA (**h**) and 10 ms. (right) Frequency distribution histograms showing the number of discrete CF-EPSC steps. Data were sampled from five control and five clodronate-treated mice in each of **g** and **h**. CF synapse elimination was significantly impaired in mice treated with clodronate at P6 (**g**, $p < 0.001$, Mann–Whitney $U$ test), but not at P11 (**h**, $p = 0.290$). Scale bars, 300 μm (**a**, **d**) and 50 μm (**b**, **c**, **e**, **f**). ***$p < 0.001$; n.s., $p > 0.05$

microglia (Fig. 2d–f). These results confirmed the local ablation of microglia by injection of clodronate liposomes.

Acute cerebellar slices were prepared from control or clodronate liposome-injected mice at P21–P31, when circuit refinement in the cerebellar cortex was almost completed. CF-mediated excitatory postsynaptic currents (CF-EPSCs) were recorded from PCs in liposome-injected lobules (Fig. 2g). In nearly 85% of PCs in control liposome-injected mice, CF-EPSCs were elicited in an all-or-nothing manner when the stimulus intensity was gradually increased (Fig. 2g, control). Conversely, in nearly 50% of PCs in the clodronate liposome-treated cerebellum, CF-EPSCs were elicited in a stepwise manner with increases in the stimulus intensity (Fig. 2g, clodronate). Because the number of CF-EPSC steps represents the number of CFs innervating a given PC, these results suggest that ablation of cerebellar microglia from P6–P8 severely impairs CF elimination. The kinetics of CF-EPSCs in mono-innervated PCs were not different between clodronate and control liposome-treated mice (Supplementary Table 1).

To examine the critical period for the clodronate-induced impairment of CF elimination, liposomal clodronate was injected at P11. Immunostaining confirmed that Iba1-positive microglia were successfully reduced at P13 (Supplementary Fig. 2) but CF elimination was not impaired in these mice (Fig. 2h). These results suggest that the critical period for microglia-dependent CF elimination was already closed by P11–P13.

We next examined the effect of pharmacological activation of microglia on CF synapse elimination. Lipopolysaccharide (LPS, 0.3 or 1.5 μg g$^{-1}$ body weight) or PBS was injected intraperitoneally at P7–P11 or P12–P16, and CF-EPSC steps were analyzed electrophysiologically after P21 (Supplementary Fig. 3). LPS administration at both postnatal periods effectively changed the morphology of the microglia (Supplementary Fig. 3a–f). However, there was no significant difference in the number of CF-EPSC steps in PCs of PBS-, 0.3 μg g$^{-1}$ LPS-, or 1.5 μg g$^{-1}$ LPS-treated mice for either administration period (Supplementary Fig. 3g, h). These results suggest that LPS-induced activation of microglia does not affect elimination of surplus CFs. Basal

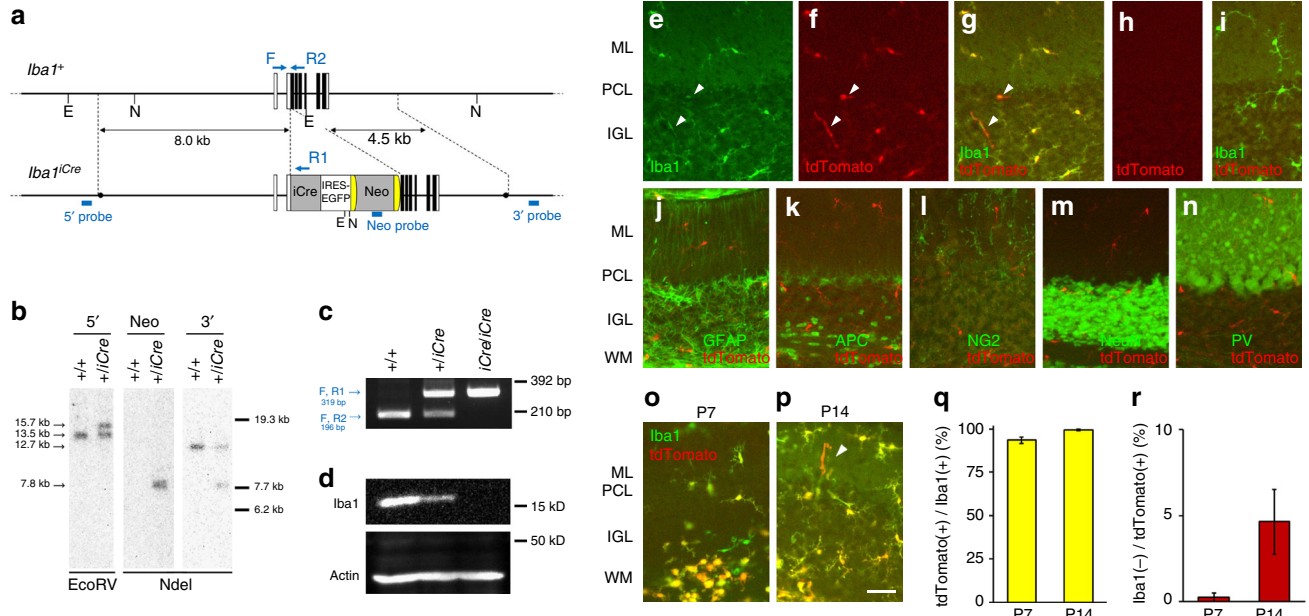

**Fig. 3** Generation of the Iba1-iCre mouse line. **a** Schema of Iba1 genomic DNA and the targeted genome. Open and filled boxes indicate noncoding and coding exons, respectively. Filled circles in the Iba1^iCre allele delineate the 5′ and 3′ termini of the targeting vector. The vector was constructed to insert an improved Cre recombinase gene (iCre), followed by an internal ribosome entry site (IRES) and a FLAG-tagged EGFP, in frame into the translational initiation site of the Iba1 gene. Blue arrows show primer positions for PCR genotyping. Blue bars indicate probes for Southern blot analysis. Two frt sequences (semicircles) are attached to remove the neomycin resistance gene (Neo). E, EcoRV; N, NdeI. **b** Southern blot analysis for genomic DNAs from wild-type (+/+) and targeted (+/iCre) ES cells. Left, EcoRV-digested genomic DNA hybridized with 5′ probe; middle and right, NdeI-digested DNA hybridized with Neo or 3′ probe, respectively. **c** Genomic PCR genotyping of Iba1^+/iCre intercrosses using primers F, R1 and R2. **d** Immunoblots of crude fractions from whole mouse brains with anti-Iba1 and actin antibodies. **e–i** Cre recombinase activity induced by Iba1-iCre in CAG-floxed STOP tdTomato reporter mice. No EGFP fluorescence was detected in the Iba1-iCre cerebellum. Immunostaining for Iba1 (green) and tdTomato fluorescence (red) in lobules IV–V in the vermis of Iba1^iCre/+; tdTomato (**e–g**) and control (Iba1^+/+; tdTomato) (**h, i**) cerebella at P30. All Iba1-positive cells expressed tdTomato, but there were several tdTomato-labeled processes with very weak or no Iba1 immunoreactivity (arrowheads in **e–p**). **j–n** Immunostaining for cell markers (green) in Iba1^iCre/+; tdTomato mice at P30; tdTomato did not overlap with markers for astrocyte (GFAP, **j**), oligodendrocyte (APC, **k**), NG2 chondroitin sulfate proteoglycan (**l**), neurons (NeuN, **m**) or inhibitory neurons (parvalbumin, **n**). **o–r** Localization of Iba1 and tdTomato signals at P7 (**o**) or P14 (**p**). **q, r** Fractions of tdTomato-labeled cells relative to Iba1-positive cells (**q**) or Iba1-negative cells relative to tdTomato-labeled cells (**r**) (n = 310 cells from two mice at P7, n = 191 cells from two mice at P14). A scale bar, 50 μm (**e–p**). Error bars, SEM

activity of microglia may be sufficient to promote CF elimination in the developing cerebellum.

**Conditional deletion of Csf1r in microglia**. Clodronate liposome administration successfully deleted microglia from the cerebellum. However, we cannot completely rule out the possibility that undetectable damage to other off-target cells with phagocytic activity, such as astrocytes, caused impairment of CF elimination in those experiments. Therefore, we next more selectively ablated microglia using cell-selective genetic manipulation. Previous reports demonstrated that CSF1R is a critical molecule for differentiation and survival of microglia in the brain[34,35]. To ablate microglia from developing cerebellum, we conditionally deleted Csf1r from microglia by crossing floxed Csf1r mice and Iba1-iCre knock-in mice to generate Csf1r(lox/lox); Iba1(iCre/iCre) mice, hereafter referred as Csf1r-cKO mice. Littermates without Cre recombinase (Csf1r(lox/lox); Iba1(wt/wt)) were used as controls.

First, we generated the Iba1-iCre mouse and examined the expression pattern of an iCre in the developing cerebellum (Fig. 3). Distribution of iCre-expressing cells was analyzed using a reporter mouse line (Iba1(iCre/+); CAG-floxed STOP tdTomato), which was produced by intercrossing the Iba1-iCre line with the Cre-inducible tdTomato reporter mouse line (CAG-floxed STOP tdTomato, Supplementary Fig. 4). TdTomato was selectively expressed in microglia (Fig. 3e–i) but not in neurons and other types of glial cells (astrocytes, oligodendrocytes, or NG2-positive cells) (Fig. 3j–n). TdTomato was already expressed at P7 in about 95% of Iba1-positive cells (Fig. 3o, q). At P14, there was a minor population of Iba1-negative but td Tomato-positive cells that associated with some blood vessels (Fig. 3p, r) in the cerebellar cortex.

The Csf1r-cKO mice grew to adulthood and were fertile. Their brain structures, the midline crossing of callosal axons and cortical thickness were largely normal, but ventricular size was enlarged at P35 (Supplementary Fig. 5). Overall foliation of the cerebellum was normal in the Csf1r-cKO mice (Supplementary Fig. 6a, b, e, f). In addition, the thickness of the granule cell layer in Csf1r-cKO mice was not significantly different from that in control mice (Supplementary Fig. 6c, g, r). The thickness of the ML was significantly increased in Csf1r-cKO mice, but the absolute difference was slight (Supplementary Fig. 6b, f, q). The density of somata of PCs, interneurons in the ML, or Golgi cells in the IGL was also normal (Supplementary Fig. 6d, h, i, m, s-u). These results indicate that the overall structure of the cerebellum is normal in Csf1r-cKO mice.

Next, we assessed reduction of microglia in the cerebellum with an antibody for CX3CR1, because Iba1 was completely deleted in Csf1r-cKO mice. In control mice, Iba1-positive microglia in the developing cerebellum colocalized with the CX3CR1 signals (Fig. 4a–i). However, the density of CX3CR1-positive microglia was severely reduced at P7 in Csf1r-cKO mice (Fig. 4j, o). Their

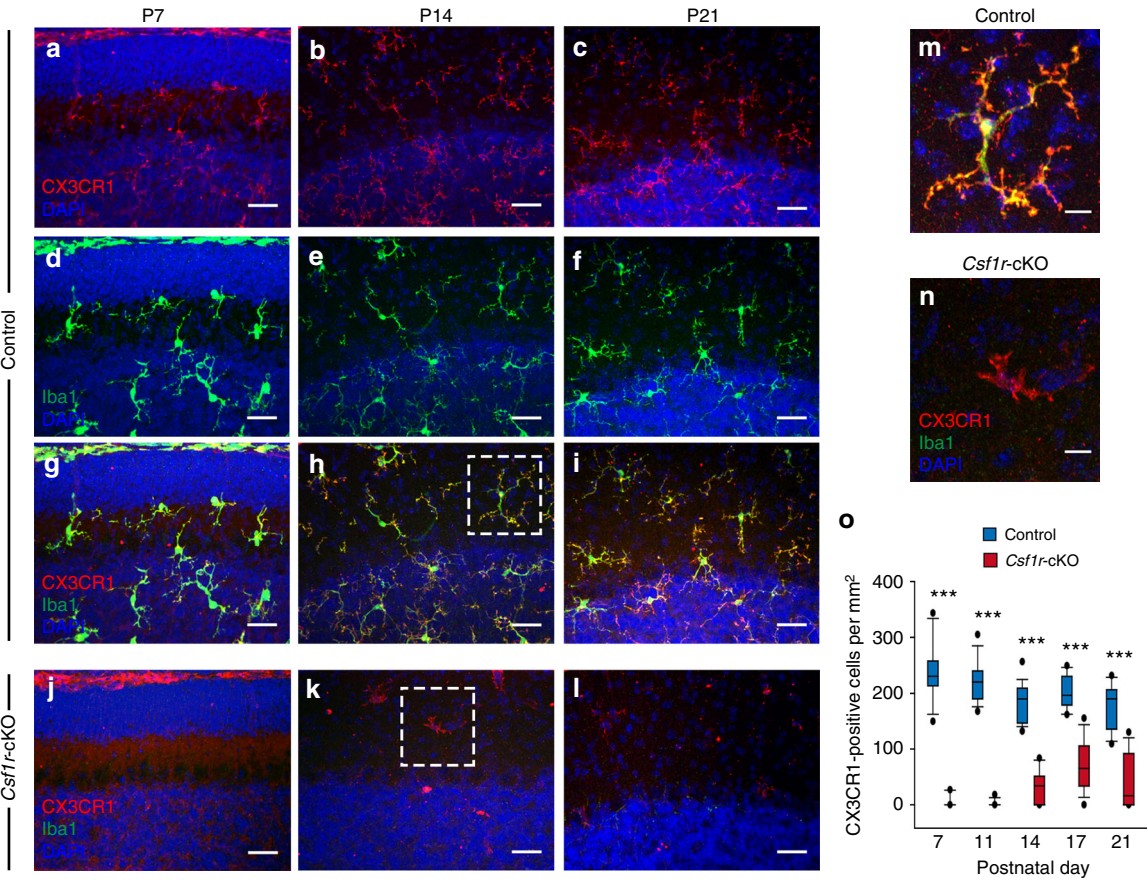

**Fig. 4** Microglia-selective deletion of *Csf1r* results in significant loss of microglia in the developing cerebellum. **a–l** Immunostaining for CX3CR1 (red), Iba1 (green), and DAPI (blue) in control mice (**a–i**) and *Csf1r*-cKO mice (**j–l**) at P7 (left), P14 (middle), and P21 (right). Images are maximum *z*-projections of slices with thicknesses of 50 μm taken from lobules VI–V in the vermis. In *Csf1r*-cKO mice, CX3CR1-positive cells were almost absent at P7 (**j**), and sparse at P14 (**k**) and P21 (**l**). **m, n** Enlargements of the CX3CR1-positive cells surrounded by dashed squares in **h** and **k**, respectively. **o** Summary box plots showing postnatal changes of the density of CX3CR1-positive cells in the IGL and ML in control (blue) and *Csf1r*-cKO (red) mice. The density of CX3CR1-positive microglia was significantly lower in *Csf1r*-cKO mice than control mice at all postnatal days that we examined ($p < 0.001$ at P7, $p < 0.001$ at P11, $p = 4.21 \times 10^{-12}$ at P14, $p = 2.01 \times 10^{-10}$ at P17, $p = 1.26 \times 10^{-7}$ at P21). ***$p < 0.001$. The density of microglia was measured in 11–19 images containing the IGL and the ML in lobules IV–V of the cerebellar vermis (256 μm × 256 μm) from three to five mice. Scale bars, 30 μm (**a–l**) and 10 μm (**m, n**)

density slightly increased after P14, but was still <30% of control mice (Fig. 4k, l, o). CX3CR1-positive cells observed after P14 in *Csf1r*-cKO mice were ameboid, possessing larger cell bodies and thicker, shorter and less ramified processes than controls (Fig. 4m, n), which was consistent with observations in *Csf1r*-null KO mice[35]. Meanwhile, expression patterns of an astrocyte marker (glial fibrillary acidic protein, GFAP), a Bergmann glia marker (3-phosphoglycerate dehydrogenase, 3PGDH) and a myelinating oligodendrocyte marker (myelin basic protein, MBP) in *Csf1r*-cKO mice were identical to those in control mice at P22–P24 (Supplementary Fig. 6j–l, n–p). Taken together, these results suggest that microglia are selectively and severely deleted from the *Csf1r*-cKO cerebellum during early postnatal development.

**CF elimination after P10–P12 is impaired in *Csf1r*-cKO mice.** Using *Csf1r*-cKO mice, we examined the functional roles of microglia in the refinement of CFs. In young adult control mice (P21–P40), nearly 80% of PCs showed CF-EPSCs with a single step. In contrast, numbers of PCs with multiple CF-EPSC steps were significantly increased in *Csf1r*-cKO mice, suggesting that developmental regression of CFs was impaired (Fig. 5a). CF-EPSCs recorded in *Csf1r*-cKO mice showed slightly larger amplitudes and faster rise times, but the paired-pulse ratio and

decay time constant were normal (Supplementary Table 2). The disparity index and disparity ratio, values that represent strengthening of the most predominant CF relative to other weaker ones, were normal in *Csf1r*-cKO mice (Supplementary Table 2). We checked the effect of *Iba1* deletion on CF elimination using *Iba1(iCre/iCre)* mice (Iba1-KO, Fig. 5a). The density of microglia in *Iba1-iCre* mice was not significantly different from that in control mice at P21 (Supplementary Fig. 7), and CF elimination was normal in these mice (Fig. 5a). The distribution histograms for *Csf1r*-cKO mice at P60–P80 (Fig. 5b) were virtually identical to those of young adult mice (P21–P41, Fig. 5a), suggesting that the impairment of CF elimination persisted into adulthood.

Next, we examined the developmental course of CF elimination in *Csf1r*-cKO mice. At P6–P8, most PCs were innervated by three or more CFs in both *Csf1r*-cKO and control mice (Fig. 5c), which suggests that formation of CF synapses was normal in *Csf1r*-cKO mice. From P10–P12, the impairment of CF synapse elimination started to become significant (Fig. 5d). CF synapse elimination progressed weakly from P13–P15 to P16–P18, even in the *Csf1r*-cKO mice, but stopped without reaching the control level (Fig. 5e, f). These results indicate that microglia-dependent CF elimination starts from P10 to P12. This postnatal period is almost coincident

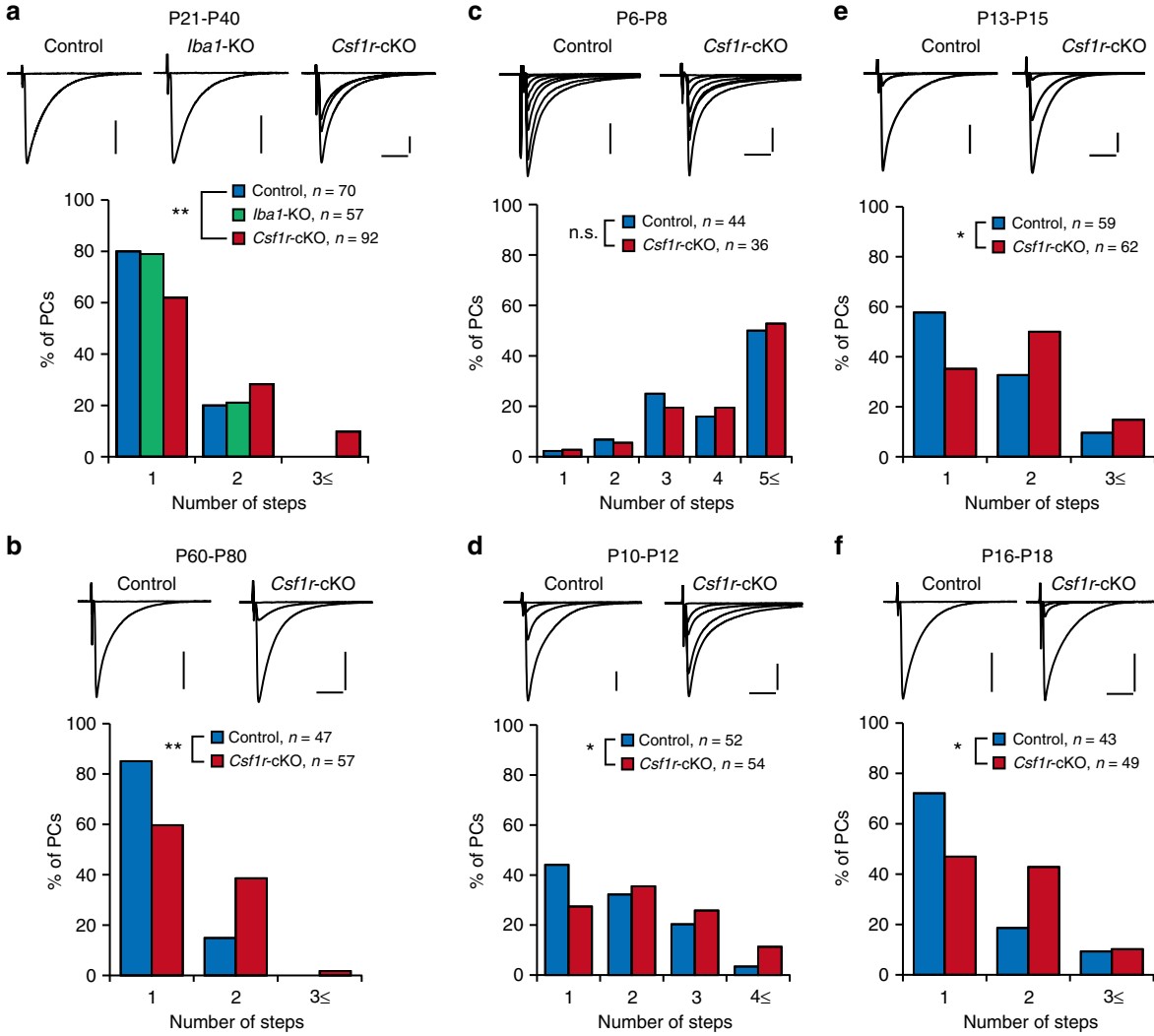

**Fig. 5** CF synapse elimination after P10–P12 is impaired in *Csf1r*-cKO mice. **a** (upper) Representative traces of CF-EPSCs in control (left), *Iba1*-KO (middle), and *Csf1r*-cKO (right) PCs in response to electrical stimulation with gradually increasing stimulus intensities; Vh = −10 mV. (lower) Frequency distribution histograms showing the number of discrete CF-EPSC steps in control (blue), *Iba1*-KO (green), and *Csf1r*-cKO (red) mice aged P21–P40. PCs were evenly sampled from all lobules. Data are sampled from four control, four *Iba1*-KO, and five *Csf1r*-cKO mice. CF synapse elimination was normal in *Iba1*-KO mice ($p = 0.887$, Mann–Whitney $U$ test) but significantly impaired in *Csf1r*-cKO mice ($p = 0.007$). **b** Frequency distribution histograms of the number of CFs in control (blue) and *Csf1r*-cKO (red) mice at P60–P80 ($p = 0.004$). **c–f** Developmental course of CF synapse elimination. Frequency distribution histograms of the number of CFs at P6–P8 (**c**, $p = 0.207$), P10–P12 (**d**, $p = 0.034$), P13–P15 (**e**, $p = 0.028$), and P16–P18 (**f**, $p = 0.030$). Data were obtained from 4 to 5 control and 4 to 5 *Csf1r*-cKO mice at each postnatal period. Scale bars, 1 nA and 10 ms. Vh = −70 mV (**c**) and −10 mV (**b, d**–**f**). *$p < 0.05$; **$p < 0.01$; n.s., $p > 0.05$

with the critical period identified from the experiments using liposomal clodronate (Fig. 2) if the delay of its pharmacological effect is taken into account.

**Microglia rarely engulf CFs during postnatal development**. We next investigated how microglia promoted CF synapse elimination. Microglia are thought to refine developing neuronal circuits by engulfing surplus synaptic structures[19–23]. Therefore, we assessed the microglial engulfment of CFs by double staining of microglia and CFs. CFs were anterogradely labeled with Alexa Fluor 568-conjugated dextran, and microglia were immunostained with Iba1 antibody in C57BL/6 mice at P10–P12. In the present study, labeled CF fragments that were completely internalized in the Iba1-labeled microglial cytoplasm were regarded as engulfed CFs. However, CF varicosities connected to the main CFs with fine processes were not regarded as engulfed, no matter

how closely associated they were. Some Iba1-labeled microglial processes were closely adjacent to Alexa Fluor 568-labeled CFs around PC somata (arrows, arrowheads, and double arrowheads in Fig. 6a–f). However, these belonged to passing CFs or collateral boutons emerging from other CF branches, suggesting that they were not completely internalized in microglia (Fig. 6b). We quantified a proportion of the microglia with inclusions of isolated CF fragments in C57BL/6 mice. For this analysis, only microglia that associated with labeled CFs in the PCL or ML were analyzed. We found only 2 out of 47 microglia (three mice) with inclusion of CF fragments (Fig. 6g). All inclusions were observed in the soma, not in the processes of microglia. These data do not completely rule out the possibility of engulfment of CFs but suggest that the incidence is very rare, even if microglia are closely associated with labeled CFs.

The Alexa Fluor 568-conjugated dextran is thought to be relatively resistant to lysosomal activity[36], but it might be rapidly

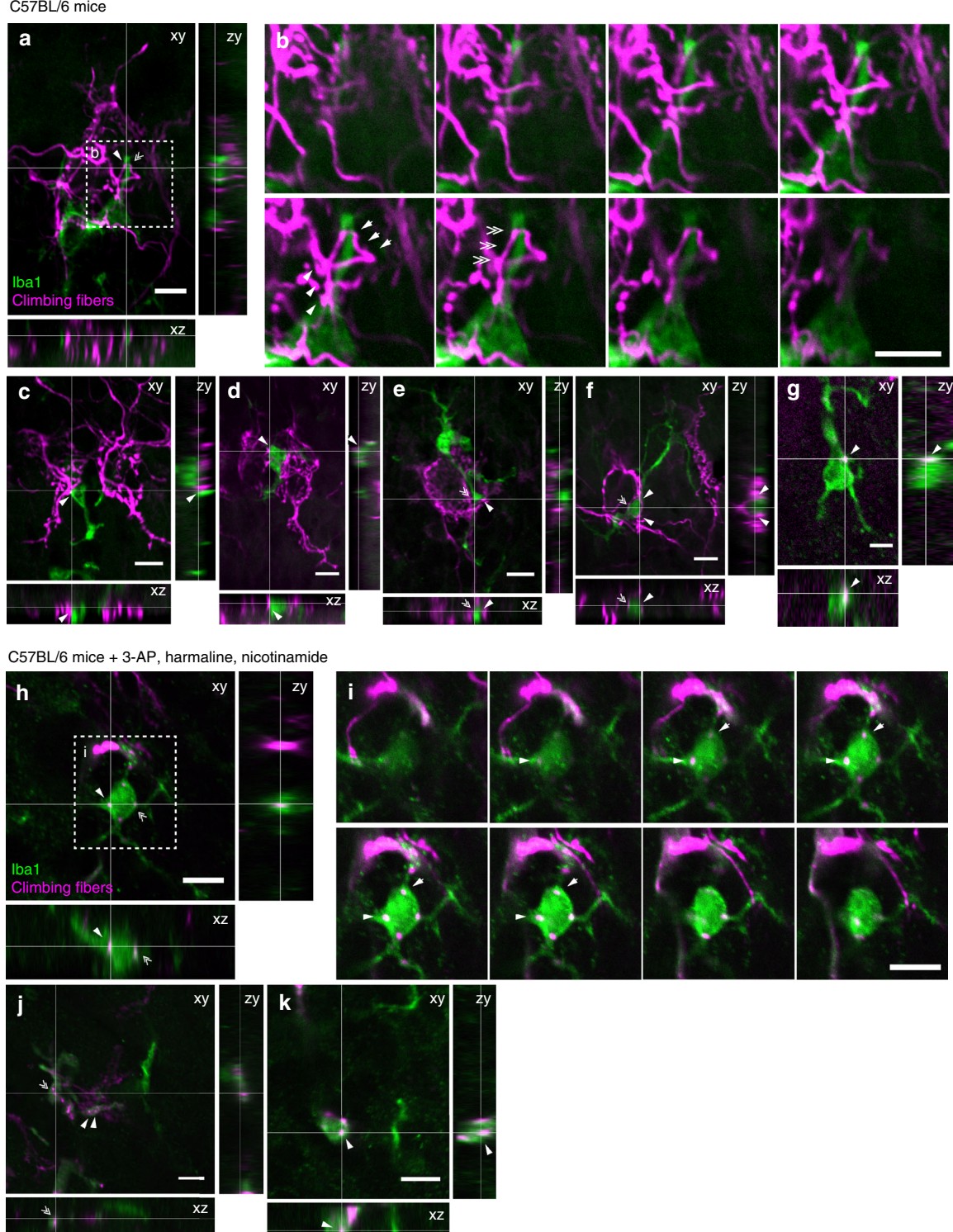

**Fig. 6** Double staining of CFs and microglia in the developing cerebellum. **a** Orthogonal fluorescence images of Iba1-positive microglia (green) and CFs anterogradely labeled with Alexa Fluor 568-dextran (magenta) in C57BL/6 mice. The *xy*-image is a single plane with its position represented as white lines in the *xz* and *zy* images. **b** Serial enlarged images ($\Delta z = 1\,\mu m$, from the left top to the right bottom) of a region surrounded by a dotted line in (**a**). Alexa Fluor 568-labeled CFs were very closely associated with Iba1-positive microglial processes but were not internalized in microglia (arrows, arrowheads, and double arrowheads). **c–f** Orthogonal fluorescence images of other microglia. **g** One of the rare examples in which an isolated CF fragment was internalized in Iba1-positive microglia. **h** Orthogonal fluorescence images similar to **a** but taken from C57BL/6 mice administrated 3-AP, harmaline and nicotinamide. **i** Serial enlarged images of a region surrounded by a dotted line in **h**. **j**, **k** Orthogonal fluorescence images of other 3-AP treated microglia. Fragments of Alexa Fluor 568-labeled CFs are included in Iba1-positive microglia (arrows and arrowheads in **h–k**), indicating that the fluorescence of Alexa Fluor 568 can be detected after being engulfed by microglia. Scale bars, 10 μm

degraded and become undetectable in our system. To test this possibility, Alexa Fluor 568-labeled CFs were partially degenerated by sequential administration of 3-acetylpyridine (3-AP), harmaline, and nicotinamide to mice[37–39]. As an analog of nicotinamide, 3-AP severely damages inferior olive (IO) neurons projecting CFs by metabolic disturbance. Co-administration of nicotinamide with 3-AP protects the rest of the central nervous system from extensive lesions[38,40,41]. Harmaline produces over-activation of the IO neurons[38,41], which apparently accelerates the metabolic changes in this nucleus. Co-administration of 3-AP with nicotinamide and harmaline locally restricts the central lesioned area to the IO[38]. In these drug-treated mice, Alexa Fluor 568-labeled CF fragments that were clearly isolated from labeled CFs and internalized in microglia (Fig. 6i) were frequently observed (25 of 33 microglia) (arrows and arrowheads in Fig. 6h–k), which indicates that Alexa Fluor 568 could be detected after being engulfed by microglia. Taken together, these results suggest that majority of surplus CFs are not removed by direct engulfment by microglia in the developing cerebellum.

**PF–PC synapses are normal in *Csf1r*-cKO mice**. We further explored mechanisms of microglia-dependent CF synapse elimination. It has been well-established that proper formation of PF–PC synapses and activation of the mGluR1-mediated signaling cascade at synapses are crucial for CF synapse elimination[15,28]. Therefore, microglia may indirectly affect CF elimination by promoting PF–PC synapse formation and/or function. To test this possibility, we examined the generation of functional PF–PC synapses at P16–P18 in *Csf1r*-cKO mice. The paired-pulse ratio (Fig. 7a) of PF-EPSCs was not different between *Csf1r*-cKO and control mice, which suggests that the presynaptic functions of PFs are normal in *Csf1r*-cKO mice. To examine the density of functional PF synapses, we recorded miniature excitatory postsynaptic currents (mEPSCs). Frequency and amplitude of mEPSCs with 10–90% rise times longer than 1 ms, which mainly arise from PF synapses[42], were analyzed. We found that the amplitude and frequency of mEPSCs in *Csf1r*-cKO mice were not significantly different from those in control mice (Fig. 7d–f). Moreover, the stimulus–response curve of PF-EPSCs was measured. Amplitudes of PF-EPSCs increased with stimulus intensity similarly in control and *Csf1r*-cKO PCs (Fig. 7b). These results suggest that the density of functional PF–PC synapses is normal in *Csf1r*-cKO mice. This was also morphologically confirmed by co-immunostaining for vesicular glutamate transporter 1 (VGluT1), a marker of PF terminals, and calbindin, a marker for PCs (Fig. 7g–l). VGluT1-positive puncta were closely associated with calbindin-labeled PC spines in both control and *Csfr1*-cKO mice (Fig. 7g–l), and the overall signal intensity of VGluT1 in the ML was unchanged in the *Csf1r*-cKO mice (Supplementary Fig. 8g, j, i).

Because activation of mGluR1 signaling at PF synapses is crucial for the late-phase CF synapse elimination[15], we next assessed abnormalities in mGluR1 signaling at PF synapses. Stimulus–response curves (Fig. 7c) and half-width of mGluR1-mediated PF-EPSCs evoked by tetanic stimulus of PFs were identical between control and *Csf1r*-cKO mice (Control: 290.8 ± 26.0 ms, $n = 17$ cells from six mice; *Csf1r*-cKO: 268.2 ± 27.6 ms, $n = 14$ cells from four mice, $p = 0.557$ by $t$ test). Immunofluorescence of mGluR1α was co-localized with that of calbindin at spines in both control and *Csf1r*-cKO mice (Supplementary Fig. 8a–f). Signal intensities of mGluR1α immunofluorescence in the ML were also identical between control and *Csf1r*-cKO mice (Supplementary Fig. 8h, k, l). Taken together, these results indicate that formation of PF–PC synapses and activation of

mGluR1-mediated signaling are not altered in *Csf1r*-cKO mice. The impairment of CF elimination in the *Csf1r*-cKO mice was therefore not a result of abnormalities in the formation or function of GC–PF–PC synapses.

**Inhibitory transmission is reduced in *Csf1r*-cKO mice**. In addition to the PF–PC synapses, we have previously demonstrated that formation and activation of GABAergic inhibitory synapses on PCs during the second postnatal week are also crucial for CF synapse elimination[29]. Therefore, it was possible that microglia affect CF elimination by promoting inhibitory synapse formation on PCs. To test this possibility, we examined miniature inhibitory postsynaptic currents (mIPSCs) and found that generation of mIPSCs was severely impaired at P10–P12 in *Csf1r*-cKO mice (Fig. 8). Cumulative frequency distributions of mIPSC amplitudes significantly shifted to lower amplitudes in the *Csf1r*-cKO mice compared with control mice (Fig. 8a–c). The average amplitude of mIPSCs was significantly smaller and their frequency was significantly lower in *Csf1r*-cKO mice than in control mice (Fig. 8d, e). Consistent with alterations in mIPSCs, signal intensity of the vesicular GABA transporter (VGAT), a marker of inhibitory synaptic terminals, was significantly decreased in the PCL, the GC, and the ML in *Csf1r*-cKO mice (Fig. 8f–j). However, the density of VGAT-positive puncta in the PCL was not changed in the *Csf1r*-cKO mice (Control: 0.024 ± 0.005 μm$^{-3}$, $n = 3$ mice; *Csf1r*-cKO: 0.020 ± 0.001 μm$^{-3}$, $n = 4$ mice, $p = 0.234$ by $t$ test), suggesting that morphogenesis of inhibitory synapses is largely normal but inhibitory synaptic transmission is impaired in *Csf1r*-cKO mice. These results indicate that microglia are crucial for the formation of functional inhibitory synapses on PCs during postnatal cerebellar development.

**Diazepam restores CF synapse elimination in *Csf1r*-cKO mice**. The results described above raise the possibility that impairment of CF elimination is caused by reduction of inhibitory synaptic transmission on PCs[29]. To test this possibility, we examined whether the administration of diazepam, a sensitizer of GABA$_A$ receptors, restored CF synapse elimination (Fig. 9). We previously demonstrated that impairment of CF synapse elimination in *Gad67*-defective heterozygous mice was restored by administration of diazepam, which suggests that developmental abnormalities caused by decreased GABA release can be recovered by the augmentation of postsynaptic GABA$_A$ receptors[29,43]. We first confirmed that acute application of diazepam (1 μM) to slices prepared from *Csf1r*-cKO mice increased the amplitude of mIPSCs (Fig. 9a–d). Diazepam or vehicle solution was injected intraperitoneally into *Csf1r*-cKO mice once per day from P9 to P12. The frequency distribution histograms of CF numbers in vehicle-treated *Csf1r*-cKO mice was not different from that of naïve *Csf1r*-cKO mice (Fig. 5a; Fig. 9e–g). Importantly, the number of PCs innervated by a single CF was significantly increased in diazepam-treated *Csf1r*-cKO mice (Fig. 9g). This result indicates that impairment of CF synapse elimination in *Csf1r*-cKO mice is mainly caused by impaired inhibitory synaptic transmission to PCs during P9–P12.

**Discussion**
As shown herein and reported previously[2–4,7], microglia are mainly distributed in the WM in the cerebellum at birth and they extensively translocate to the cerebellar cortex after P7, mainly from the WM (Fig. 1)[4]. In this study, we found that microglia transiently show biased distribution from the PCL to the IGL that peaked beneath the PCL at P8–P9. This postnatal period is coincident with the start of CF synapse elimination in the mouse cerebellum[33], suggesting the contribution of microglia to this

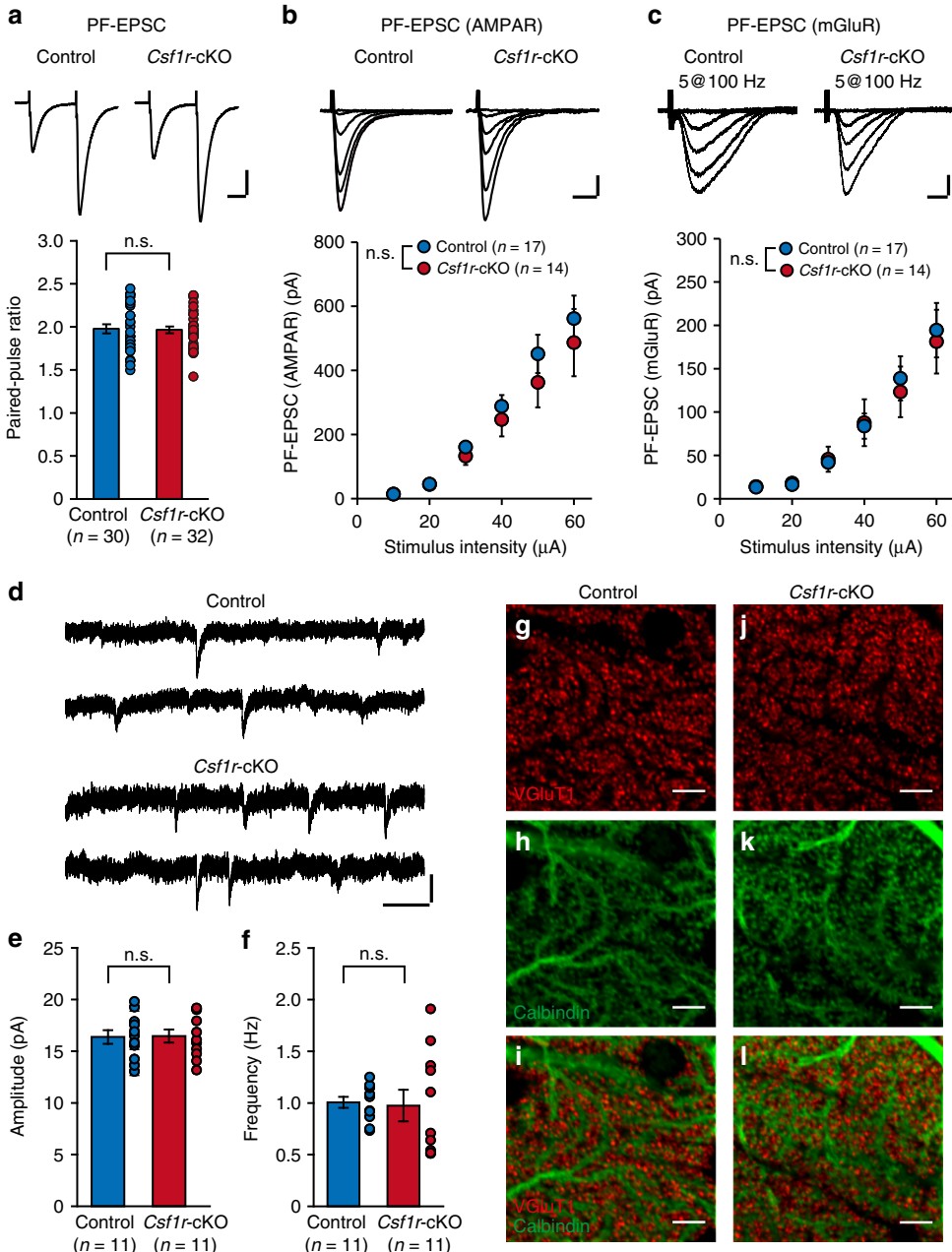

**Fig. 7** Parallel fibers to PC synapses are normal in *Csf1r*-cKO mice. **a** (upper) PF-EPSCs recorded in a control or a *Csf1r*-cKO PC in response to paired stimuli with an interval of 50 ms. (lower) The paired-pulse ratio was not different between control ($n = 30$ cells from seven mice) and *Csf1r*-cKO ($n = 32$ cells from eight mice, $p = 0.972$, Mann–Whitney $U$ test). **b** (upper) PF-EPSCs evoked by stimulus intensities of 10, 20, 30, 40, 50, and 60 μA in a control or a *Csf1r*-cKO PC. (lower) Stimulus–response curve of PF-EPSCs. The peak amplitudes of PF-EPSCs were measured and plotted against stimulus intensity. No difference is observed between control ($n = 17$ cells from six mice) and *Csf1r*-cKO ($n = 14$ cells from four mice, $p = 0.459$, two-way RM ANOVA). **c** (upper) Traces of mGluR1-mediated currents evoked by tetanic stimulation (five pulses at 100 Hz) with 10, 20, 30, 40, 50, and 60 μA stimulus intensities. (lower) The stimulus–response curves of mGluR1-mediated currents. These data were obtained from the same cells shown in (**b**). No difference was observed ($p = 0.886$, two-way RM ANOVA). **d** Miniature EPSCs (mEPSCs) recorded from a control or a *Csf1r*-cKO PC in the presence of 1 μM TTX and 100 μM picrotoxin. Amplitude (**e**, $p = 0.921$, $t$ test) or frequency (**f**, $p = 0.555$, Mann–Whitney $U$ test) did not differ between control ($n = 11$ cells from three mice) and *Csf1r*-cKO ($n = 11$ cells from three mice) mice. Vh = −70 mV (**a-f**). Electrophysiological data were obtained from PCs in lobules IV–V at P16–P18 (**a-f**). Scale bars, 100 pA and 20 ms (**a, c**), 200 pA and 10 ms (**b**), and 10 pA and 200 ms (**d**). n.s., $p > 0.05$. **g-l** Immunostaining for VGluT1 (**g, j**), calbindin (**h, k**), and merged of them (**i, l**). Images were taken from the ML in lobules IV–V in the vermis. Scale bars, 5 μm. PC dendrites possessed dense spines that faced VGluT1 puncta in both control and *Csf1r*-cKO mice. Data are presented as mean ± SEM

event. In fact, this period was temporally coincident with the time window when clodronate liposome injection was effective (Fig. 2) and when the impairment of synapse elimination in *Csf1r*-cKO began (Fig. 5). These lines of evidence suggest that initiation of microglia-dependent cerebellar circuit maturation has a close

relationship with the transient accumulation of microglia around the PCL–IGL area.

Microglia are absent from the CNS in *Csf1r*-null KO mice[1,34,35], and pharmacological inhibition of CSF1R clears ~99% of microglia from the brain[44]. Therefore, it is widely believed that

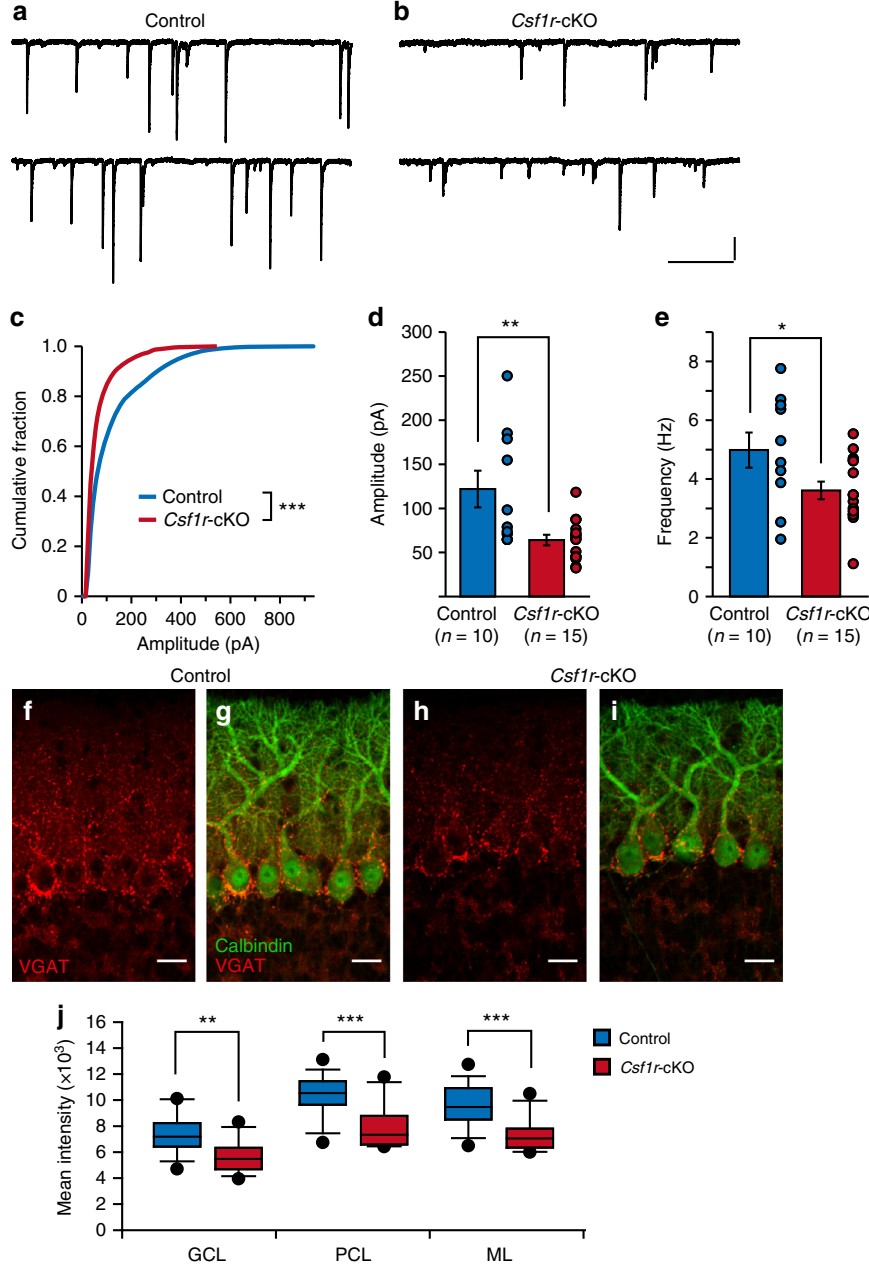

**Fig. 8** Generation of functional inhibitory synapses is impaired in *Csf1r*-cKO mice. **a**, **b** Representative traces of mIPSCs recorded in a control (**a**) or a *Csf1r*-cKO (**b**) PC in the presence of 1 μM TTX and 10 μM NBQX; Vh = −70 mV. Scale bars, 100 pA and 500 ms. **c** Cumulative fractions of amplitudes of mIPSCs in control (blue) and *Csf1r*-cKO (red) mice. Recordings were obtained from PCs in lobules IV–V at P10–P12. The incidence of mIPSCs with smaller amplitudes is significantly increased in *Csf1r*-cKO mice ($p < 0.001$, Kolmogorov–Smirnov test). Graphs consist of 8426 events from 10 control cells from three mice and 9847 events from 15 *Csf1r*-cKO cells from six mice. **d**, **e** Averaged amplitude (**d**) and frequency (**e**) of mIPSCs. Amplitude was significantly smaller (**d**, $p = 0.006$, Mann–Whitney *U* test) and frequency was significantly lower (**e**, $p = 0.013$) in *Csf1r*-cKO (red) than in control (blue) mice. Data were obtained from 10 cells from three control mice and 15 cells from six *Csf1r*-cKO mice. Data are presented as mean ± SEM. **f–i** Immunostaining for VGAT (**f**, **h**), and merged images for VGAT (red) and calbindin (green) (**g, i**) in a control (**f, g**) and a *Csf1r*-cKO mouse (**h, i**) at P11. Images were taken from lobules IV–V in the vermis. Scale bars, 20 μm. **j** Mean fluorescence intensity of VGAT staining in the granule cell layer (GCL), the PC layer (PCL), and the molecular layer (ML) in lobules IV–V of cerebellar vermis. Signal intensity of VGAT was significantly lower in *Csf1r*-cKO mice than in control mice in all layers (GCL: $p = 0.001$; PCL: $p < 0.001$; ML: $p < 0.001$, *t* test). Each data set was obtained from 15 images (256 μm × 256 μm) containing GCL, PCL, and ML from three control and three *Csf1r*-cKO mice at P10–P12. Scale bars, 30 μm. *$p < 0.05$; **$p < 0.01$; ***$p < 0.001$

CSF1R is crucial for the proliferation, differentiation, and survival of microglia. CSF1R is mainly expressed in microglia in the brain, but some immunohistochemical analyses reported that it is also expressed in nestin-positive neural progenitor cells and subsets of neurons, including PCs[45–47]. Therefore, we generated microglia-selective conditional KO mice by crossing *Csf1r*-floxed and *Iba1*-*iCre* mice to abolish microglia from the brain. *Csf1r*-null KO mice rarely survive beyond P21 and exhibit severe abnormalities in brain architecture, such as a small brain size, enlarged ventricles, a thinned cortex, defects in midline crossing of callosal axons, and a loss of microglia[34, 35]. At least some of these abnormalities in brain structures are likely caused by the deletion of *Csf1r* from

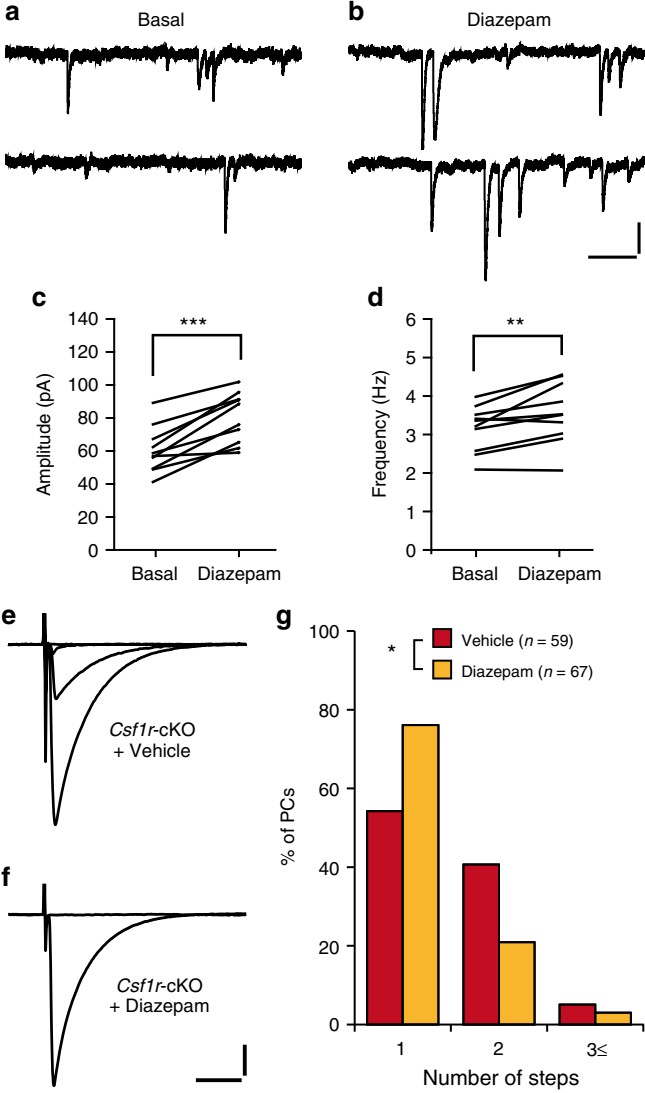

**Fig. 9** Administration of diazepam restores CF elimination in *Csf1r*-cKO mice. **a–d** Representative traces of mIPSCs recorded from a *Csf1r*-cKO PC before (**a**) and after (**b**) application of 1 µM diazepam. Vh = −70 mV. Scale bars, 30 pA and 250 ms. Mean amplitude of mIPSCs significantly increased during bath application of diazepam (**c**, $p < 0.001$, paired $t$ test, $n = 10$ cells from three mice). Frequency was also significantly increased by the diazepam application (**d**, $p = 0.006$). **e**, **f** Representative traces of CF-EPSCs in *Csf1r*-cKO mice treated with vehicle (**e**) or diazepam (**f**); Vh = −10 mV. Scale bars, 500 pA and 10 ms. **g** Frequency distribution histograms showing the number of CFs innervating each PC in vehicle- and diazepam-treated *Csf1r*-cKO mice. Data were sampled from five vehicle-treated and five diazepam-treated mice. The distribution of the histogram of vehicle-treated *Csf1r*-cKO mice is virtually identical to that of naïve *Csf1r*-cKO mice ($p = 0.540$, Mann–Whitney $U$ test). The percentage of mono-innervated PCs is significantly increased in diazepam-treated *Csf1r*-cKO mice ($p = 0.011$, Mann–Whitney $U$ test). $*p < 0.05$; $**p < 0.01$; $***p < 0.001$

non-microglial cells, because similar abnormalities are partially reproduced in mice in which *Csf1r* is conditionally deleted from nestin-positive neural progenitor cells[45]. Our analysis also supports these data. *Csf1r*-cKO mice showed severe reduction in microglial numbers in the developing cerebellum (Fig. 4), which confirms that *Csf1r* in microglia is crucial for survival of microglia in the brain. However, *Csf1r*-cKO mice survived to adulthood and were fertile. The abnormalities in brain architecture reported in *Csf1r*-null mice were not observed, except for the enlarged

ventricular size (Supplementary Fig. 5). Taken together, CSF1R signaling in microglia and non-microglial cells likely has distinct roles in the formation and maintenance of brain architecture. In the *Csf1r*-cKO mice, *Csf1r* would also be deleted from peripheral macrophages[48]. Because peripheral immune responses are thought to modulate gene expression and synaptic functions in the brain[49], deletion of *Csf1r* in peripheral macrophages might affect CF synapse elimination. However, impairment of CF synapse elimination was reproduced by deletion of microglia using intracerebellar injection of liposomal clodronate (Fig. 2), indicating that microglia within the cerebellum play the dominant role in circuit refinement.

Our electrophysiological and morphological analyses suggested that GABAergic transmission on PCs was significantly impaired in *Csf1r*-cKO mice (Fig. 8). Molecular mechanisms for microglia-dependent facilitation of development of inhibitory synapses in the cerebellum are currently unknown but brain-derived neurotrophic factor (BDNF)-tyrosine receptor kinase B (TrkB) signaling is a candidate. BDNF is produced by and released from microglia[18], the amplitude and frequency of mIPSCs in PCs are reduced in *TrkB*-KO mice[50], and the number of GAD65-positive boutons is reduced in cerebellum-specific *TrkB*-KO mice[51]. Importantly, CF synapse elimination is also impaired in mutant mice with defective BDNF–TrkB signaling[50,52,53]. Cytokines might also be candidates. Interleukin-1β (IL-1 β) enhances plasma membrane insertion of GABA$_A$ receptors in cultured neurons and oocytes[54], and in vivo nanoinjection of tumor necrosis factor α (TNFα) induces GABA$_A$ receptor trafficking to synaptic membranes in spinal cord neurons[55]. Identification of microglia-derived factors regulating inhibitory synaptic transmission on PCs will need to be identified in future studies.

CF synapse elimination during postnatal development is finely regulated by neuron–neuron interactions[15,33]. Many previous reports have demonstrated that CF synapse elimination is regulated by PF–PC synaptic activities. CF synapse elimination is impaired in animals that have abnormalities in the generation of PF–PC synapses[15,26,33]. PF synapses are thought to have roles in restricting the CF innervation sites to the proximal dendrites and in activating the mGluR1 signaling cascade crucial for late-phase CF elimination. In addition, CF elimination is also regulated by interneuron–PC interactions[29]. CF elimination is impaired in *Gad67* heterozygous KO mice and this impairment is reversed by diazepam[29]. Activation of GABAergic synapses is thought to regulate CF synapse elimination, presumably by affecting intracellular calcium transients associating with CF inputs[29,56]. In the present study, we found that formation of functional PF synapses and activation of the mGluR1 signaling cascade were normal, but inhibitory synaptic transmission was severely impaired in the *Csf1r*-cKO mice. The impairment of CF synapse elimination in *Csf1r*-cKO mice was recovered by administration of diazepam, which suggests that the defect in GABAergic inhibition to PCs causes impairment of CF synapse elimination. In *Csf1r*-cKO mice, regression of CF synapses was impaired from P10 to P12. It weakly proceeded until P14 but stopped thereafter (Fig. 5). This developmental course of CF regression in *Csf1r*-cKO mice was very similar to that observed in *Gad67* heterozygous KO mice[29]. These lines of evidence suggest that microglia indirectly affect CF synapse elimination by regulating GABAergic inhibition in the developing cerebellum. This notion is also supported by the rare observation of CFs internalized in microglia in the developing cerebellum (Fig. 6), which suggests that CF synapses are not generally removed by direct engulfment by microglia. However, because activated microglia have been reported to induce axonal retraction by expressing contact-dependent repulsive axon guidance molecules[57,58], it remains a possibility that microglia also

induce CF retraction by mechanisms not dependent on engulfment.

In conclusion, our present study revealed that microglia primarily facilitate formation of functional inhibitory synapses in the developing cerebellum, thereby promoting elimination of surplus CFs by driving mechanisms inherent in PCs. Microglia–neuron and neuron–neuron interactions cooperatively participate in postnatal circuit refinement in the cerebellum.

## Methods

**Animals**. All experiments were conducted according to the guidelines of the experimental animal ethics committees (A16-137, M-P12-138, SA00041) and the biosafety committee for living modified organisms (28-223, 70, SD00800) of Hiroshima University, the University of Tokyo, and Niigata University, respectively. Male and female C57BL/6, *Csf1r(lox/lox); Iba1(iCre/iCre), Csf1r(lox/lox); Iba1 (+/+), Iba1(iCre/+); CAG-floxed STOP tdTomato* and *Iba1(iCre/iCre)* mice aged from P5 to P80 were used for experiments. We generated *Iba1-iCre* (Fig. 3a–c) and *CAG-floxed STOP tdTomato* (Supplementary Fig. 4) mice. To generate *Csf1r(lox/lox); Iba1(iCre/iCre)* mice, we mated *Csf1r*-floxed mice (Jackson Laboratories, stock no. 021212)[59] with *Iba1-iCre* mice. Genotypes of mice were examined by PCR of genomic DNA extracted from their tails. The following specific primers were used for *Iba1-iCre* mice: F, 5′-GGTGTCAGCAGAAGCTGATG-3′; R1, 5′-AGCATCTTCCAGGTGTGTTC-3′; R2, 5′-CCTTCCCTACCCTGCAAATC-3′. We maintained all mice in specific pathogen-free conditions on a reversed 12 h light/dark cycle (lights off at 8 PM) with free access to food and water.

**Immunohistochemistry**. Mice were deeply anesthetized by intraperitoneal injection of pentobarbital (100 mg kg$^{-1}$ body weight) and transcardially fixed with 2 or 4% paraformaldehyde. Brains were removed, postfixed in the same fixative at 4 °C overnight, and stored in phosphate-buffered saline (PBS) containing 0.1% NaN$_3$. Parasagittal cerebellar sections of 50 μm thickness were prepared from the vermis with a vibratome slicer (TDK-1000, Dosaka EM). The slices were incubated in 10% normal donkey serum, then in primary antibodies overnight at 4 °C. Primary antibodies used in the present study were rabbit anti-Iba1 (1:500, #019-19741, Wako), mouse anti-CX3CR1 (1:100, #149002, Biolegend), rabbit or mouse anti-NeuN (1:500, #ab177487, Abcam; 1:1000, #MAB377, Merck), mouse anti-GFAP (1:500, #MAB3402, clone GA5, Merck; 1:250, #3670, Cell Signaling Technology), rabbit anti-NG2 (1:200, #AB5320, Merck), chicken anti-myelin basic protein (1:100, #AB9348, Merck), mouse anti-APC (1:250, #NB600-1021, Novus Biologicals), rabbit anti-neurogranin (1:200, #AB5620, Merck), goat or rabbit anti-calbindin (1:200, #Calbindin-Go-Af1040 or #Calbindin-Rb-Se-1, Frontier Institute), rabbit anti-VGAT (1:200, #VGAT-Rb-Af500, Frontier Institute), goat or guinea pig anti-parvalbumin (1:200, #PV-Go-Af460; 1:500, #PV-GP-Af1000, Frontier Institute), rabbit anti-mGluR1α (1:200, #mGluR1a-Rb-Af811, Frontier Institute), and goat anti-VGluT1 (1:200, #VGluT1-Go-Af310, Frontier Institute). After incubating in the primary antibodies, sections were washed and then incubated in Alexa Fluor 488-, Alexa Fluor 555-, Alexa Fluor 555- (1:500, Thermo Fisher Scientific, MA) or Alexa Fluor 488-, DyLight 405- or Cy3- (1:200, Jackson Immunoresearch) conjugated, species-specific secondary antibodies for 2 h at room temperature. Counter staining was performed with DAPI (1 μg ml$^{-1}$, Thermo Fisher Scientific) or Neuro Trace 530/615 Red or 500/525 Green Fluorescent Nissl Stain (1:500, Thermo Fisher Scientific). We took images with a confocal laser scanning microscope (LSM700, Zeiss) or a fluorescent microscope (IX71, OLYMPUS) of immunostained slices from lobules IV–V because of their long straight alignment of cerebellar layers. We analyzed the images with ImageJ software (Rasband, W.S., ImageJ, U. S. National Institutes of Health, Bethesda, Maryland, USA, https://imagej.nih.gov/ij/, 1997–2016). When taking images of large regions, a KEYENCE BZ-X700 fluorescence microscope (KEYENCE) was used.

**Production of transparent brain slices (SeeDB2)**. Mice were deeply anesthetized with pentobarbital (100 mg kg$^{-1}$ body weight) and brains were fixed as described above. Parasagittal cerebellar sections of 500 μm thickness were prepared from the vermis. The sections were cleared according to the SeeDB2 protocol[32]. Briefly, the sections were incubated with 2% saponin in PBS overnight. After immunohistochemistry against Iba1 and calbindin, the sections were incubated serially with 33, 50, and 90% Omnipaque 350 (Daiichi Sankyo) in water containing 2% saponin for 6–12 h each. Then the sections of lobules IV–V were immersed in 100% Omnipaque 350 and imaged with a confocal laser scanning microscope (FV1000, Olympus) equipped with 25× objective lens (XLPLN25XIMA, Olympus).

**Electrophysiology**. Mice aged P6–P80 were deeply anesthetized by CO$_2$ inhalation and decapitated. The brains were quickly removed, and parasagittal slices with thickness of 250 μm were prepared from cerebellar vermis with a vibratome slicer (VT1200S, Leica) in chilled normal artificial cerebrospinal fluid (ACSF) containing 125 mM NaCl, 2.5 mM KCl, 2 mM CaCl$_2$, 1 mM MgSO$_4$, 1.25 mM NaH$_2$PO$_4$,

26 mM NaHCO$_3$, and 20 mM glucose, bubbled with 95% O$_2$ and 5% CO$_2$. After slice preparation, slices were kept at 25 °C in normal ACSF.

Whole-cell recordings were made from visually identified PCs using an upright microscope (BX50WI, Olympus). PCs were equally sampled from all lobules except the experiment shown in Fig. 2, in which PCs within the liposome-injected lobule were selectively examined. All data were recorded at 32 °C with an EPC10 patch clamp amplifier (Harvard Bioscience) without correction of liquid junction potential. Online data acquisition and offline data analysis were performed using Fit Master software (Harvard Bioscience) and the Mini Analysis Program (Version 6.0.7, Synaptosoft). The pipette solution for voltage clamp recordings was composed of 60 mM CsCl, 10 mM Cs$_D$-gluconate, 20 mM TEA-Cl, 20 mM BAPTA, 4 mM MgCl$_2$, 4 mM ATP, 0.4 mM GTP, and 30 mM HEPES (pH 7.3, adjusted with CsOH). For recording of CF- or PF-EPSCs, picrotoxin (100 μM, Tocris) was added to normal ACSF to block inhibitory synaptic transmission. For recording of mIPSCs, TTX (1 μM, Wako) and NBQX (10 μM, Tocris) were added to the normal ACSF. For recording of stimulus–response curves of PF-EPSCs, the pipette solution was composed of 65 mM K-gluconate, 65 mM Cs-methanesulfonate, 10 mM KCl, 5 mM sucrose, 0.4 mM EGTA, 20 mM HEPES, 1 mM MgCl$_2$, 4 mM ATP, and 1 mM GTP (pH 7.3, adjusted with CsOH). Stimulation pipettes were filled with normal ACSF and placed on the ML. Square pulses were applied for focal stimulation (duration, 0.1 ms; amplitude, from 0 to 90 V). To elicit mGluR1-mediated currents, PFs were stimulated five times at 100 Hz in the presence of NBQX (20 μM) and picrotoxin (100 μM). CFs were repetitively stimulated at 0.2 Hz by an electrode placed in the IGL around the PC soma. The stimulation pipette was systematically moved around the PC soma. The number of CFs innervating the recorded PC was estimated as the number of discrete CF-EPSC steps during gradually increasing stimulus intensity. Approximately half of the mice in each experimental group were examined and analyzed blinded for their genotypes or experimental operations.

**Anterograde labeling of CFs**. Under anesthesia with isoflurane, a glass pipette filled with a solution of Alexa Fluor 568-conjugated dextran (10% in PBS, 10,000 MW; Thermo Fisher Scientific) was stereotaxically administered to the IO of C57BL/6 mice at P7–P9. The tracer was injected iontophoretically with pulse current (10 μA, 700 ms) at 0.5 Hz for 20 min. After tracer injection, the wound was stitched with silk sutures, and mice were moved to a cage with a heating lamp. After recovery from anesthesia, mice were moved to their home cage. After 2–3 days recovery, mice were deeply anesthetized with pentobarbital (100 mg kg$^{-1}$ body weight) and transcardially fixed with 4% PFA for immunohistochemical experiments. Images were taken using the same procedure described in the immunostaining section above. We omitted mice from the analysis if the Iba1-positive cells on the surface of the cerebellar cortex (Fig. 1a) were heavily stained by Alexa Fluor 568-conjugated dextran that leaked from the injection sites in the medulla. In such mice, microglia with Alexa Fluor 568 signals were very frequently observed in the cerebellum, probably because some of the stained microglia on the surface of the brain migrated into the developing cerebellum[4].

In some experiments (Fig. 5g, h), 3-AP (75 μg g$^{-1}$ body weight, Sigma-Aldrich), harmaline (15 μg g$^{-1}$ body weight, Sigma-Aldrich), and nicotinamide (300 μg g$^{-1}$ body weight, Sigma-Aldrich) were injected intraperitoneally at P9, after tracer injection at P7, to induce degeneration of CFs. Harmaline and nicotinamide were administered 3 and 4.5 h after 3-AP, respectively. In these experiments, mice were fixed at P14 for histological assessment.

**Intracerebellar liposome application**. C57BL/6 mice at P6–P7 or P10–P11 were anesthetized with isoflurane, and the occipital bone and the dura over cerebellar lobules VI–VIII were carefully removed. A volume of ~200 nl of control or clodronate liposomes (#160-0364-1 or #160-0432-1, Katayama Chemical) was injected over 20 min into lobules VI, VII, or VIII of the cerebellar vermis at a depth of 200–250 μm using a glass capillary and a microinjector (NANOJECT II, Drummond Scientific, or UMP3 ultra micro pump, WPI). After tracer injection, the wound was stitched with silk sutures, and mice were moved to a cage with a heating lamp. After recovery from anesthesia, mice were moved to their home cage with free access to food and water. Innervation by CFs in the injected lobule was examined electrophysiologically at P21–P31.

**Intraperitoneal drug applications**. LPS (from *E. coli* O111: B4; Sigma-Aldrich) was intraperitoneally injected into C57BL/6 mice once per day from P7 to P11 or P12 to P16. The doses of LPS were set to 0.3 and 1.5 μg g$^{-1}$ of body weight, and the same volume of PBS was injected as a control. Innervation by CFs was examined electrophysiologically at P21–P32.

Diazepam (10 μg g$^{-1}$ of body weight; Sigma-Aldrich) was administered to *Csf1r*-cKO mice once per day from P9 to P12. As a control, vehicle solution, which was composed of 40% propylene glycol, 10% ethanol, and 50% distilled water, was administered to a different age-matched group of *Csf1r*-cKO mice. Innervation by CFs was examined electrophysiologically at P25–P41.

**Statistical analysis**. No statistical method was used to decide sample sizes, but they were chosen based on previous studies with similar methodologies. No data, except for Fig. 6 (the detail is explained in Anterograde labeling of CFs section in

the Methods), were excluded from the analysis. For all pharmacological experiments, mice were randomly chosen from amongst littermates for the drug or vehicle injections. Throughout the text and figures, $n$ represents the number of PCs analyzed unless otherwise stated in the figure legends. Averaged values were represented as mean ± SEM. In graphs for electrophysiological experiments, bar graphs representing mean values and the individual data points are displayed in parallel. In boxplots, center lines, whiskers, dots, and bonds of box represent the median, the 10th/90th, the 5th/95th, and the 25th/75th percentiles, respectively. Statistical significance was assessed by two-sided $t$ test or the Mann–Whitney $U$ test, depending on whether the data sets passed the normality test and equal variance test, unless otherwise stated in the text. Statistical analyses were conducted with SigmaPlot 12.1 (Systat Software) and $p$ values smaller than 0.001 were described as $p < 0.001$, otherwise actual values were described in the text. Differences between two samples were considered statistically significant if the $p$ value was less than 0.05.

**Data availability**. The authors declare that the main data supporting the findings of this study are available within the article and its Supplementary Information files. Data that support the findings of this study are available from the corresponding author upon reasonable request.

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

## Acknowledgements

We thank M. Mitsushima, M. Nonaka, M. Takata, R. Takimoto, H. Teramichi, K. Yamaoka, S. Suzuki, and R. Kubo for their assistance in preliminary experiments, and M. Kawamura for technical assistance. This work was supported by Grants-in Aid for Scientific Research (15K183420A and 17K070580A to H.N., 26350979 to M.A., 17H01387 to S.O., 25000015, 17H03551, and 18H04947 to K.H., 25117006 to S.O. and K.H.) from the Ministry of Education, Culture, Sports, Science and Technology of Japan. This work was also supported by the Strategic Research Program for Brain Sciences (17dm0107093h0002) from AMED to K.H. We thank Ann Turnley, Ph.D., from Edanz Group (www.edanzediting.com/ac) for editing a draft of this manuscript.

## Author contributions

H.N. and K.H. designed the study. H.N., M.A., C.M., and T.I. performed experiments. H.N., M.A., C.M., T.I., and K.H. analyzed the data. M.A., T.I., S.O., and K.S. contributed reagents/analysis tools. H.N., M.A., C.M., T.I. S.O., K.S., and K.H. wrote the paper.

## Additional information

**Competing interests:** The authors declare no competing interests.

