## [Peer Review File · Nature Communications]

Reviewers' comments:

Reviewer #1 (Remarks to the Author):

This is a very interesting study that demonstrates the role of microglia in mediating synaptic pruning of climbing fibers from Purkinje cells, a classic model of synapse elimination in the CNS. The authors use pharmacologic and genetic means to ablate microglia and demonstrate, using immunocytochemistry and electrophysiology, impairment in the transition from multiple CF innervation to single innervation. Further, the authors make the interesting finding that loss of microglia results in impaired inhibitory innervation of PC and that enhancement of inhibitory signaling restores the CF elimination in the microglia ablated mice. How microglia shape neural circuits in health and disease is an important topic making this a very significant study that will be of general interest to the scientific community. However, there are some methodological concerns that need to be addressed.

Major:

Iba-cre mice that are used to ablate microglia are described in another manuscript that is in preparation. Some characterization of these mice is necessary before this manuscript can be considered.

There is no characterization of the Iba-KO mice- is there an effect on microglia?
Show CX3CR1 immunostaining to ascertain that microglia are intact.

The analysis of the immunostaining with VGluT1 and mGluR1 in figure 6 is not convincing and is certainly not described well. Even within the presented images there is variability in the intensity and it is not clear which regions were analyzed and whether a subtle change would be picked up with such an approach. Analysis of higher magnification images showing unchanged density of PC spines contacted by VGluT1 would be more convincing as well as demonstrating unchanged mEPSCs.

Although generally the study seems to be appropriately powered, in some experiments (i.e. pharmacological deletion of microglia) the number of mice used is insufficient (n=3).

It is not always clear what is the N used for statistical analysis (i.e. Figure 7D, E) and in some cases the data sets seem to be artificially inflated by considering cells or images as N.

Minor:

It is not stated which part of the cerebellum was used for immunostaining or recordings (vermis?).

Fig 3m- control instead of conttol.

Reviewer #2 (Remarks to the Author):

MANUSCRIPT #: NCOMMS-17-28798-T

TITLE: Microglia-dependent GABAergic synaptogenesis promotes refinement of climbing fibers to Purkinje cell synapses

AUTHORS: Prof Hashimoto and co-workers

In this paper, the authors used a conditional genetic approach in mouse to test the role of microglia during cerebellar development. They found that conditional deletion of *Csf1r* using an *Iba1-Cre* mouse line resulted in cerebellar phenotypes that included structural and functional defects. Specifically, they found that loss of *Csf1r* resulted in climbing fiber pruning problems, and using pharmacology, immunohistochemistry and electrophysiology they demonstrate that the microglia impact climbing fiber elimination by modulating the function of inhibitory synapses during development. These data were supported by a decrease in *Vgat* expression in the molecular layer interneurons.

This is a very interesting manuscript that tackles a very exciting area of cerebellar biology - development of climbing fiber to Purkinje cell mono-innervation. There has been much work conducted on climbing fiber pruning from several major groups such as Kano, Nishiyama, and others. However, this is the first full demonstration of a role for microglia during the process of climbing fiber poly- to mono-innervation on Purkinje cells. The data that are presented here are overall very convincing and the writing of the text is excellent. I have only one major problem with this piece of work. I discuss it below in addition to a number of other concerns that should be addressed.

1. The major problem with this manuscript is that the authors do not provide any information on the *Iba1-Cre* mouse. They cite the line as "in preparation". Although I like the paper, without a proper validation of the *Iba1-Cre* mouse in the context of this work, I cannot properly evaluate the strength of the deletion and its consequences on the cerebellum. A full description must be provided including documentation of expression after crossing to a reporter mouse line, reporter expression at various developmental and adult ages, recombination efficiency of the line, *Iba1-Cre* reporter expression compared to endogenous *Iba1* expression, reporter expression in cerebellum and other parts of the brain, reporter expression (or lack thereof) in neuronal lineages, etc. As I am sure the authors appreciate, this has to be provided in order for a reader to have confidence that the deletion of *Csf1r* is robust and specific.

Additional concerns:

2. The Title should be revised to something along the lines of "Microglia-dependent GABAergic synaptogenesis promotes refinement of climbing fiber to Purkinje cell innervation"

3. The specific Cre line (*Iba1*) must be mentioned in the abstract. The current description that is provided is much too vague.

4. The authors stated "Therefore, the functional roles of microglia likely alter with the progress..." Please consider switching the word "alter" to "change" or "adapt". Alter is a poor choice of word.

5. In the sentence "...inhibitory synaptic transmissions were attenuated..." I don't see the need for plural. Please change to singular.

6. The authors state "Immunostaining confirmed that Iba1-positive microglia were successfully reduced at P13 (data not shown)..." Why is the data not shown? This is important!

7. The authors report "These results suggest that the critical period for microglia-dependent CF elimination is around P7–P11." Why not inject the liposomal clodronate even earlier? That way you could properly define the time window. As it stands, your lower cutoff at P7 seems rather arbitrary.

8. The authors conclude "These results suggest that overactivation of microglia does not affect elimination of surplus CFs." Okay, but the data could also suggest that your loss of function pharmacology induces additional off target damage. A fair and balanced conclusion must be made.

9. The authors state "and Iba1-Cre knock-in mice (Abe et al, in preparation)..." As I mentioned before, for a high impact journal the reader must have access to the details otherwise one cannot fairly judge or appreciate the manipulation.

10. The authors performed a characterization of glial number and structure in the Csf1r-cKO cerebellum. However, no equivalent study was performed for the neurons. Reasonable markers are available for Purkinje cells, granule cells, Golgi cells and unipolar brush cells – cell number and distribution can be easily examined. A deeper analysis of neuronal architecture and some aspects of circuit connectivity must be provided. The Vglut1 and 2 staining is a good start, but more is required.

11. Along the same lines as above, sagittal images through the developing and adult cerebellum of control and KO mice would be useful. Does the overall structure of the cerebellum (i.e. lobule patterning and lobule shape) really develop normally? It is currently very hard to tell.

12. The authors state "that impairment of CF elimination is caused by hypoplasia of inhibitory synapses..." Hypoplasia is not an accurate description. Please change your choice of wording.

13. The authors conclude "our present study revealed that microglia directly and indirectly regulate..." What is your evidence for direct and indirect? I don't understand what you mean.

14. The harmaline and related experiments were poorly justified. I do not understand what

the motivation was for using these specific drugs. Please provide a more concrete rationale, including specifically what each compound does to climbing fiber function, and specifically in the developing mice. What are you actually doing to the circuit? And what are the effects on glia as well. Again, please discuss all the details.

15. It is not so clear to me how many cells were recorded in each experiment. The number of mice and the number of events is stated. But what about individual neurons in each paradigm?

16. I would also strongly suggest that for the electrophysiology graphs the individual data points should be shown on the bar graph – the scatter and grouping of cells is very valuable when analyzing this type of data. Without this the variability and error bars are hard to interpret.

17. Apologies if I have missed it, but is the Vgat expression intensity decreased evenly throughout the cerebellar cortex? What about in the deep nuclei? This type of comparison is important because it helps provide specificity and support for your hypothesis.

18. Along the same lines as above, was the number of Vgat immunoreactive terminals decreased? Please provide the data because even if it is not, it helps put the decreased expression data into a more understandable context.

Reviewer #3 (Remarks to the Author):

Contribution of microglia to functional neural circuit formation is one of the hottest topics of current neuroscience. In this work, Nakayama et al. investigated how microglia contribute to climbing fiber synapse elimination in the developing cerebellum, which is a representative model of neural circuit refinement of the developing nervous system. They found that developmental climbing fiber synapse elimination was impaired when cerebellar microglia were ablated by local injection of clodronate liposome into the cerebellum at P6. In contrast, local injection of clodronate liposome at P11 had no effect on climbing fiber innervation, suggesting that the critical period for microglia-dependent climbing fiber elimination is around P7 to P11. They also generated mice with microglia-specific deletion of colony stimulating factor 1 receptor (*Csf1r*) gene (*Csf1r* cKO mice) in which microglia were mostly ablated in the developing brain. In *Csf1r* cKO mice, climbing fiber synapse elimination after P10 was impaired and this impairment persisted into adulthood. Importantly, while some microglial processes were closely adjacent to climbing fibers in wild-type mice at P10–P12, climbing fiber terminals that were engulfed by microglia were not observed. Formation and function of parallel fiber to Purkinje cell synapses were normal in *Csf1r* cKO mice. However, the amplitude and frequency of mIPSCs recorded in Purkinje cells and intensity of VGAT immunoreactivity were significantly reduced in the cerebellum of *Csf1r* cKO mice. Enhancing GABA_A receptor sensitivity by diazepam rescued the impairment of climbing fiber synapse elimination of *Csf1r* cKO mice. The authors conclude that microglia facilitates climbing fiber synapse elimination not by directly engulfing redundant climbing fiber synapses but by promoting formation of GABAergic innervation to Purkinje cells around P7 to P11 and

thereby permitting the inhibitory interneuron-dependent mechanisms of climbing fiber elimination.

This work has disclosed a novel mechanism how microglia contributes to neural circuit refinement during postnatal development. The experiments have been carefully performed and the data are generally in very high quality. I have several comments which the authors may consider to improve their manuscript.

Major points

(1) Page 13, line 236-248, Fig. 5:

The sample images shown in Figure 5 are fine and nicely represent the authors' arguments. However, I feel some quantification is necessary to substantiate their claim that microglia does not seem to engulf redundant climbing fiber synapses during postnatal development.

(2) Page 14 line 268 – page 15, line 270, Fig. 6c:

In the specimen records of mGluR1-mediated slow EPSCs, the time course looks clearly shorter in *Csf1r*-cKO than in Control. Is the difference statistically significant if the data from populations of cells are compared.

(3) Page 16, line 305 – page 17, line 308:

The authors should show that diazepam actually enhances mIPSCs in acute cerebellar slices from *Csf1*-cKO mice.

(4) Although the manuscript is generally clearly written, there are a number of awkward expressions and grammatical errors. I feel rigorous rewriting with the help of native English speakers/writers is necessary to improve this scientifically strong paper.

Minor points

(1) The title does not seem to properly reflect the content of the manuscript. I suggest an alternative title, "Microglia permit climbing fiber synapse elimination by promoting GABAergic synaptogenesis in the developing cerebellum".

(2) The abstract does not seem to summarize the manuscript well. The authors should mention that there is no evidence that microglia directly engulf redundant climbing fiber synapses.

(3) Page 7, line 121-122, Figure 1g: "This trend was also observed in analyses of the nearest-neighbor distance (Fig. 1g right)"

I do not see "similar tendency" in Fig. 1g right when compared to Fig. 1g left.

Point-by-point responses to Reviewers' comments.

Response to comments by Reviewer #1

“.....How microglia shape neural circuits in health and disease is an important topic making this a very significant study that will be of general interest to the scientific community.”

We appreciate the positive evaluation of our study provided by Reviewer #1.

Major:

1. Iba-cre mice that are used to ablate microglia are described in another manuscript that is in preparation. Some characterization of these mice is necessary before this manuscript can be considered.

[Our response to the comment]

We appreciate the reviewer's helpful suggestions. We examined Cre-expressing cells in the developing cerebellum using a reporter mouse line (Iba1(iCre/+); tdTomato/+) that was produced by intercrossing the Iba1-iCre line with the Cre-inducible tdTomato reporter mouse line (CAG-floxed STOP tdTomato). The expression of tdTomato was selectively co-localized with Iba1 but not markers for neurons or other glial cells in the young adult cerebellum. CFs were not stained therefore recombination did not occur in the inferior olive. Although there was a population of Iba1-negative but tdTomato-positive cells that associated with some blood vessels (Fig. 3p,r) in the cerebellar cortex at P14, the population was very minor relative to the unstained blood vessels. The tdTomato was expressed at P7 in approximately 95% of Iba1-positive cells, suggesting that recombination had already occurred in the majority of microglia by the first postnatal week.

We now present these data and basic information about the generation of Iba1-iCre mice in Figure 3. We also provide information in the Results section (page11, line 5-).

2. There is no characterization of the Iba-KO mice- is there an effect on microglia? Show CX3CR1 immunostaining to ascertain that microglia are intact.

[Our response to the comment]

We thank the reviewer for raising an important issue. As suggested by the

reviewer, we examined CX3CR1 immunostaining in the Iba1-KO mice. We found that their density of microglia was normal and now present and explain these data in Supplementary Figure 7 and in the Results section (page 13, line 2 from the bottom-).

3. The analysis of the immunostaining with VGluT1 and mGluR1 in figure 6 is not convincing and is certainly not described well. Even within the presented images there is variability in the intensity and it is not clear which regions were analyzed and whether a subtle change would be picked up with such an approach. Analysis of higher magnification images showing unchanged density of PC spines contacted by VGluT1 would be more convincing as well as demonstrating unchanged mEPSCs.

[Our response to the comment]

In response to the reviewer's suggestion, we tried to examine the density of VGluT1-positive puncta in the molecular layer at higher magnification. However, discrimination of individual VGluT1-positive puncta was difficult with our confocal microscope (Figure 7g, j). Therefore, we confirmed this point by analyzing mEPSCs. Miniature EPSCs with rise times slower than 1 ms, which are thought to arise from PF but not CF synapses (Yamasaki et al., 2006, Ichikawa, 2016), were analyzed. As a result, we found that the amplitude and frequency of mEPSCs were normal in Csf1r-cKO mice. Taken together with the normal input-output relationship of evoked PF-EPSCs (Figure 7b), these data confirm the normal density of functional PF terminals in Csf1r-cKO mice. In addition, we now present higher magnification images of co-immunolabeling of VGluT1 or mGluR1a with calbindin in the molecular layer in Figure 7g-l and Supplementary Figure 8. Overall, morphology of spines and their close association with VGluT1- or mGluR1a-puncta were similar between control and Csf1r-cKO mice.

We now present the results of mEPSCs on page 17, line 10- and in Figure 7d-f. The comments on the higher magnification images are presented on page 17, line 2 from the bottom-, and in Figure 7g-l for VGluT1, and in Supplementary Figure 8a-f for mGluR1.

4. Although generally the study seems to be appropriately powered, in some experiments (i.e. pharmacological deletion of microglia) the number of mice used is insufficient (n=3).

[Our response to the comment]

We have now added two more mice for the analysis of pharmacological deletion of

microglia (Figure 2). In addition, we also added two more mice to the last experiment using diazepam (Figure 9).

5. *It is not always clear what is the N used for statistical analysis (i.e. Figure 7D, E) and in some cases the data sets seem to be artificially inflated by considering cells or images as N.*

[Our response to the comment]

We appreciate the reviewer's helpful suggestions. We have now carefully revised the number of specimens.

Minor:

6. *It is not stated which part of the cerebellum was used for immunostaining or recordings (vermis?).*

[Our response to the comment]

The vermis was used for all experiments. However, analyzed lobules were different depending on the experiment. Electrophysiological analysis of clodronate-treated mice was conducted on lobules VI–VIII around the injection site. Electrophysiological analysis of Csf1r-cKO mice was conducted from all lobules. All morphological experiments were conducted in lobules IV–V because of their long, straight alignment of layers. We now provide this information in the Methods ‘**Immunohistochemistry**’, ‘**Production of transparent brain slices (SeeDB2)**’, ‘**Electrophysiology**’ and ‘**Intracerebellar liposome application**’ subsections.

7. *Fig 3m- control instead of conttol.*

[Our response to the comment]

We have corrected the typo.

Response to comments by Reviewer #2

“.....This is a very interesting manuscript that tackles a very exciting area of cerebellar biology - development of climbing fiber to Purkinje cell mono-innervation”.

We appreciate the positive evaluation of our study provided by Reviewer #2.

1. The major problem with this manuscript is that the authors do not provide any information on the Iba1-Cre mouse. They cite the line as “in preparation”. Although I like the paper, without a proper validation of the Iba1-Cre mouse in the context of this work, I cannot properly evaluate the strength of the deletion and its consequences on the cerebellum. A full description must be provided including documentation of expression after crossing to a reporter mouse line, reporter expression at various developmental and adult ages, recombination efficiency of the line, Iba1-Cre reporter expression compared to endogenous Iba1 expression, reporter expression in cerebellum and other parts of the brain, reporter expression (or lack thereof) in neuronal lineages, etc. As I am sure the authors appreciate, this has to be provided in order for a reader to have confidence that the deletion of Csf1r is robust and specific.

[Our response to the comment]

We appreciate the reviewer’s helpful suggestions. As described in the reply to the 1st comment by Reviewer #1, we have now added information about Iba1-iCre mice. Cre-expressing cells were checked using a reporter mouse line (Iba1(Cre/+); CAG-floxed STOP tdTomato). Co-immunostainings demonstrated that tdTomato was selectively expressed in microglia but not neurons or other glial cells in the developing cerebellum. TdTomato was already expressed at P7 in most microglia. We now present these data in Figure 3 and provide information in the Results section (page11, line 5-).

2. The Title should be revised to something along the lines of “Microglia-dependent GABAergic synaptogenesis promotes refinement of climbing fiber to Purkinje cell innervation”

[Our response to the comment]

We appreciate the reviewer’s helpful suggestions. We have now revised the title to “Microglia permit climbing fiber elimination by promoting GABAergic inhibition in the developing cerebellum”.

3. *The specific Cre line (Iba1) must be mentioned in the abstract. The current description that is provided is much too vague.*

[Our response to the comment]

In the revised manuscript, we added information about the Iba1-iCre mice in the Abstract (page 3, line 7-).

4. *The authors stated “Therefore, the functional roles of microglia likely alter with the progress...” Please consider switching the word “alter” to “change” or “adapt”. Alter is a poor choice of word.*

[Our response to the comment]

We have corrected the word “alter” to “change” (page 4, line 10-).

5. *In the sentence “...inhibitory synaptic transmissions were attenuated...” I don't see the need for plural. Please change to singular.*

[Our response to the comment]

We have changed this to the singular (page 6, line 6-).

6. *The authors state “Immunostaining confirmed that Iba1-positive microglia were successfully reduced at P13 (data not shown)...” Why is the data not shown? This is important!*

[Our response to the comment]

Followed the reviewer’s helpful advice we have now added this information on page 13, line 2 from the bottom- and present the data in Supplementary Figure 2.

7. *The authors report “These results suggest that the critical period for microglia-dependent CF elimination is around P7–P11.” Why not inject the liposomal clodronate even earlier? That way you could properly define the time window. As it stands, your lower cutoff at P7 seems rather arbitrary.*

[Our response to the comment]

We previously demonstrated that CF synapse elimination starts from P8 in wild-type mice (Hashimoto et al., 2007). CF innervation weakly increases until P7 but starts to decrease from P8 in mice. Thus, we planned to induce degeneration of microglia just around the start of CF synapse elimination by clodronate administration at P6. Moreover, because our morphological analysis demonstrated that most microglia were localized in the white matter and not around PCs at P5 (Figure 1), we thought that effects of microglial ablation from the cerebellar cortex would be minimal or undetectable before P6. Therefore, we did not examine liposomal-clodronate injection before P7.

We think that analysis of Csf1r-cKO mice is more appropriate for studying the influence of microglia on CF synaptogenesis and elimination before P7 than analyzing by the liposomal clodronate administration. As mentioned in the Results (page 10, line 17-) and discussed in the reply to the next comment, we cannot completely rule out the possibility of off-target damage by liposomal clodronate administration. For microinjection of liposomes, a glass microelectrode was inserted into the cerebellum and may transiently affect microglial activity. In contrast, genetic manipulation in the Csf1r-cKO mice is more microglia-specific and less invasive. Our analyses of Csf1r-cKO mice showed that CF innervation was normal at P6–P8 (fig. 5c), which suggests that initial CF innervation was normal and microglia-dependent CF refinement starts after this postnatal period.

The reviewer's helpful suggestion indicated to us that the start of the critical period was not correctly addressed in the clodronate experiment. We have now revised this comment to "These results suggest that the critical period for microglia-dependent CF elimination was already closed by P11–P13" (page 9, line 17-).

8. The authors conclude "These results suggest that overactivation of microglia does not affect elimination of surplus CFs." Okay, but the data could also suggest that your loss of function pharmacology induces additional off target damage. A fair and balanced conclusion must be made.

[Our response to the comment]

We thank the reviewer for raising an important issue. The reason for the lower effectiveness of LPS on CF elimination was unclear but basal activity of microglia in the developing cerebellum may be enough to promote CF elimination. Otherwise, it also remains a possibility that CF synapse elimination may be promoted by signaling cascades not activated by LPS in microglia.

As pointed out by the reviewer, we cannot completely rule out the possibility that

impairment of CF elimination is caused by off-target damage by liposomal clodronate. The off-target problem is a common and crucial issue for pharmacological experiments and therefore we used the more specific procedure of genetic manipulation to delete microglia. Ablation of microglia by cell-specific deletion of *Csf1r* also caused impairment of CF elimination, which confirms that microglia are crucial for CF elimination during postnatal development.

We have now added comments about the possibility of off-target damage in our pharmacological experiments at the introduction of generation of *Csf1r*-cKO mice (page 10, last line 14-). Furthermore, we changed the word “overactivation” to “LPS-induced activation” (page 10, line 8-).

9. The authors state “and Iba1-Cre knock-in mice (Abe et al, in preparation)...” As I mentioned before, for a high impact journal the reader must have access to the details otherwise one cannot fairly judge or appreciate the manipulation.

[Our response to the comment]

As described in the reply to your 1st comment, we have now added information about *Iba1*-Cre mice in Figure 3, Supplementary Figure 4 and the Results section.

10. The authors performed a characterization of glial number and structure in the Csf1r-cKO cerebellum. However, no equivalent study was performed for the neurons. Reasonable markers are available for Purkinje cells, granule cells, Golgi cells and unipolar brush cells – cell number and distribution can be easily examined. A deeper analysis of neuronal architecture and some aspects of circuit connectivity must be provided. The Vglut1 and 2 staining is a good start, but more is required.

[Our response to the comment]

Following the reviewer’s suggestion, we have now examined the influence of conditional deletion of *Csf1r* on neurons in the cerebellar cortex. We have added new data for immunostaining of neurons (NeuN, Neuro Trace), PCs (calbindin), interneurons in the molecular layer (parvalbumin) and Golgi cells (neurogranin) in *Csf1r*-cKO mice in Supplementary Figure 6. We found that densities of PCs (Supplementary Figure 6t), PV-positive interneurons (Supplementary Figure 6s) and Golgi cells (Supplementary Figure 6u) were not significantly changed in *Csf1r*-cKO mice. The data are presented at page 12, line 5-.

The thickness of the granule cell layer in the vermis was normal in *Csf1r*-cKO

mice (Supplementary Figure 6r). Moreover, analysis of the input-output relationship of the PF-EPSC suggested that the density of functional PF synapses was normal (Figure 7b). We have now examined mEPSCs and found that the amplitude and frequency were not significantly different in Csf1r-cKO mice (Figure 7d-f), which also suggests the normal density of functional PF synapses. These results indicate that the GC–PF system is normally constructed in Csf1r-cKO mice.

New data indicate that densities of the somata of inhibitory interneurons in the cerebellar cortex were normal in Csf1r-cKO mice (page12, line 5-). In addition, the density of VGAT-positive inhibitory terminals in the PCL was not significantly changed in Csf1r-cKO mice (page 19, line 15-). However, generation of mIPSCs was significantly impaired, and the intensity of immunostaining of VGAT was decreased, both in the molecular layer and the granule cell layer (fig. 8j). These results suggest that morphogenesis of inhibitory interneurons and their synapses are largely normal, but inhibitory synaptic transmission is impaired in Csf1r-cKO mice.

11. Along the same lines as above, sagittal images through the developing and adult cerebellum of control and KO mice would be useful. Does the overall structure of the cerebellum (i.e. lobule patterning and lobule shape) really develop normally? It is currently very hard to tell.

[Our response to the comment]

Following the reviewer’s suggestion we have added morphological analyses of the cerebellum. Foliation and layer structures were normal in Csf1r-cKO mice (Supplementary Figure 6a, e). As discussed in our response to the 10th comment of Reviewer 2, the thickness of the GCL was normal in Csf1r-cKO mice. The thickness of the ML was significantly increased in Csf1r-cKO mice, but the difference in the absolute values was very slight (Supplementary Figure 6q). We now present the data in Supplementary Figure 6 and describe it on page11, from the last line.

12. The authors state “that impairment of CF elimination is caused by hypoplasia of inhibitory synapses...” Hypoplasia is not an accurate description. Please change you choice of wording.

[Our response to the comment]

We have now changed “hypoplasia of inhibitory synapses” to “reduction of inhibitory synaptic transmission” (page 20, line 6-).

13. The authors conclude “our present study revealed that microglia directly and indirectly regulate...” What is your evidence for direct and indirect? I don't understand what you mean.

[Our response to the comment]

In the previous version, “directly” and “indirectly” meant roles of microglia in functional inhibitory synaptogenesis and CF elimination (that is indirectly mediated by a neuronal mechanism), respectively. However, as pointed out by the reviewer, this sentence is a little vague and we have now omitted it (page 25, second paragraph).

14. The harmaline and related experiments were poorly justified. I do not understand what the motivation was for using these specific drugs. Please provide a more concrete rationale, including specifically what each compound does to climbing fiber function, and specifically in the developing mice. What are you actually doing to the circuit? And what are the effects on glia as well. Again, please discuss all the details.

[Our response to the comment]

A procedure often used to lesion inferior olive (IO) neurons is 3-AP administration; 3-AP is an analogue of nicotinamide and functions as a metabolic antagonist, leading to decreased levels of nicotinamide and consequent inhibition of NAD⁺-dependent reactions. It is reported that sole administration of 3-AP severely damages IO neurons by metabolic disturbance, but several other brain nuclei, primarily in the medulla, are also weakly damaged (Declin and Escubi, 1974). However, co-administration of harmaline and nicotinamide can restrict the central lesioned area to the IO (Llinas et al., Science, 1975). Co-administration of nicotinamide with 3-AP protects the rest of the central nervous system from extensive lesions (Shimantov, 1976, Llinas et al., 1975, Hicks, 1955). Harmaline produces over-activation of the IO neurons (Llinas and Volkind, 1973, Simantov, 1976), which apparently accelerates the metabolic changes in this nucleus. Morphological analyses using light and electron microscopy have demonstrated that cerebellar architecture, apart from CFs, is intact after 3-AP administration (Declin, 1974, Declin and Escubi, 1974, Simantov, 1976, Sotelo, 1975). Therefore, we think that the influence on other cells is minimal.

Of course, we cannot completely rule out the possibility of morphologically undetectable damage by these drugs. However, the most important purpose of this analysis was to confirm that the fluorescence of Alexa Fluor 568 was detectable even

after being engulfed by microglia. We think that this point is correctly assessed in our analysis. We now present an explanation for the pharmacological effects of these drugs in the Results section (page 16, line 5-).

15. It is not so clear to me how many cells were recorded in each experiment. The number of mice and the number of events is stated. But what about individual neurons in each paradigm?

[Our response to the comment]

We have carefully revised the number of specimens.

16. I would also strongly suggest that for the electrophysiology graphs the individual data points should be shown on the bar graph – the scatter and grouping of cells is very valuable when analyzing this type of data. Without this the variability and error bars are hard to interpret.

[Our response to the comment]

We have followed the reviewer's advice and now present individual data points and their average in all graphs.

17. Apologies if I have missed it, but is the Vgat expression intensity decreased evenly throughout the cerebellar cortex? What about in the deep nuclei? This type of comparison is important because it helps provide specificity and support for your hypothesis.

[Our response to the comment]

As discussed in the reply to the 10th comment, we measured the VGAT signal intensity in the granule cell layer and the molecular layer, as well as the PC layer (Figure 8). As a result, we found that the VGAT signal was decreased throughout the cerebellar cortex. This result suggests that microglia promote functional inhibitory synapse formation in the overall cerebellar cortex. These data are presented at page 19, line 12-.

18. Along the same lines as above, was the number of Vgat immunoreactive terminals

decreased? Please provide the data because even if it is not, it helps put the decreased expression data into a more understandable context.

[Our response to the comment]

We appreciate the reviewer's helpful suggestions. We have now examined the density of VGAT puncta. Because inhibitory synaptogenesis around PC somata by basket cells is known to be important for CF synapse elimination (Nakayama et al., 2012), the density of VGAT puncta in the PCL was examined. We found that the density was not significantly different between control and Csf1r-cKO mice. These new data suggest that morphogenesis of inhibitory synapses is largely normal, but inhibitory synaptic transmission is impaired in CSF1r-cKO mice. We describe this result in the Results section (page 19, line 15-).

Response to comments by Reviewer #3

“.....This work has disclosed a novel mechanism how microglia contributes to neural circuit refinement during postnatal development. The experiments have been carefully performed and the data are generally in very high quality”.

We appreciate the positive evaluation of our study by Reviewer #3.

Major points

(1) Page 13, line 236-248, Fig. 5:

The sample images shown in Figure 5 are fine and nicely represent the authors' arguments. However, I feel some quantification is necessary to substantiate their claim that microglia does not seem to engulf redundant climbing fiber synapses during postnatal development.

[Our response to the comment]

We appreciate the reviewer's helpful suggestions. We quantified a proportion of microglia with inclusions of isolated CF fragments in wild-type mice. We omitted mice from the analysis if the Iba1-positive cells on the surface of the cerebellar cortex were heavily stained by Alexa Fluor 568-conjugated dextran that leaked from the injection site in the medulla. In such mice, microglia in the cerebellum were heavily stained, probably because some of these stained microglia migrated into the developing cerebellum (Quadros, 1997).

Labeled CF fragments that were completely internalized in the Iba1-labeled microglial cytoplasm were regarded as engulfed CFs. In contrast, CF varicosities connected to the main CFs with fine processes were not regarded as engulfed, no matter how closely associated they were. Furthermore, only microglia that associated with labeled CFs in the PCL or ML were analyzed. We found only 2 out of 47 microglia (3 mice) with inclusion of CF fragments. All inclusions were observed in the soma, not in the processes of microglia. On the contrary, about 76% of microglia had completely isolated and internalized CF fragments in 3AP-treated mice. These data do not completely rule out the possibility of engulfment of CFs but suggest that the incidence is very rare. We now present the data in the Results section (page 15, line 12-).

(2) Page 14 line 268 – page 15, line 270, Fig. 6c:

In the specimen records of mGluR1-mediated slow EPSCs, the time course looks clearly shorter in Csf1r-cKO than in Control. Is the difference statistically significant if the data

from populations of cells are compared.

[Our response to the comment]

We thank the reviewer for raising an important issue. We examined the half width of the mGluR-mediated current and confirmed that there was no significant difference between control and *Csf1r*-cKO mice. We now present the data on page 18, line 7-.

(3) Page 16, line 305 – page 17, line 308:

*The authors should show that diazepam actually enhances mIPSCs in acute cerebellar slices from *Csf1r*-cKO mice.*

[Our response to the comment]

We thank the reviewer for raising an important issue. We examined the acute effect of diazepam on mIPSCs recorded in PCs in *Csf1r*-cKO mice. Miniature IPSC amplitudes were actually enhanced by diazepam. We now present and explain these data in Figure 9a–d and in the Results section (page 20, lines 13-).

(4) Although the manuscript is generally clearly written, there are a number of awkward expressions and grammatical errors. I feel rigorous rewriting with the help of native English speakers/writers is necessary to improve this scientifically strong paper.

[Our response to the comment]

A native English-speaking editor has again copyedited our manuscript.

Minor points

(1) The title does not seem to properly reflect the content of the manuscript. I suggest an alternative title, “Microglia permit climbing fiber synapse elimination by promoting GABAergic synaptogenesis in the developing cerebellum”.

[Our response to the comment]

We have now revised the title to “Microglia permit climbing fiber elimination by promoting GABAergic inhibition in the developing cerebellum”.

(2) The abstract does not seem to summarize the manuscript well. The authors should mention that there is no evidence that microglia directly engulf redundant climbing fiber synapses.

[Our response to the comment]

We have now added comments in the abstract (page 3, line 8-) describing that the engulfment of CFs is negligible.

(3) Page 7, line 121-122, Figure 1g: “This trend was also observed in analyses of the nearest-neighbor distance (Fig. 1g right)”

I do not see “similar tendency” in Fig. 1g right when compared to Fig. 1g left.

[Our response to the comment]

We thank the reviewer for raising an important issue. At P8–P9, the nearest-neighbor distance tends to increase from the white matter side to the external granule cell sides. However, in this period, the nearest-neighbor distance transiently decreased beneath the Purkinje cell layer, which suggests an increase in the density of microglia. However, as suggested by reviewer, we recognize that these data describe the same conclusion as Figure 1g left. Thus, we have now omitted Figure 1g right.

REVIEWERS' COMMENTS:

Reviewer #1 (Remarks to the Author):

The authors have addressed my concerns.

The images in figure 3, characterizing the Iba1-iCre mice are of low quality (somewhat fuzzy and with high background signal). These need to be improved prior to publication so they are on par with other figures. This should be the case especially if they appear in the main manuscript rather than in supplementary material.

Reviewer #2 (Remarks to the Author):

The authors have submitted a revised manuscript of their work showing that microglia ablation alters climbing fiber to Purkinje cell innervation in mice.

The authors have provided a point by point explanation for how they addressed each comment provided by the reviewers. They have done an excellent job in addressing the comments and incorporating the changes into the paper.

I have no further concerns.

Reviewer #3 (Remarks to the Author):

The authors have satisfactorily addressed all the comments by this reviewer and also most of the concerns by raised by the other reviewers. They have improved their manuscript significantly.

Point-by-point responses to Reviewers' comments.

Response to comments by Reviewer #1

The authors have addressed my concerns.

We appreciate the positive evaluation of our study provided by Reviewer #1.

1. The images in figure 3, characterizing the Iba1-iCre mice are of low quality (somewhat fuzzy and with high background signal). These need to be improved prior to publication so they are on par with other figures. This should be the case especially if they appear in the main manuscript rather than in supplementary material.

[Our response to the comment]

We appreciate the reviewer's helpful suggestions. We have now exchanged Figure 3i and 3l with high background signals to other images. In addition, contrast of Figure 3e and 3f was slightly processed equally across the entire images. Unfortunately, images in Figure 3 were obtained using the fluorescent microscope (IX71, OLYMPUS), and thus quality was not identical to that of other images obtained using the confocal microscope (LSM700, Zeiss). However, we think that revised images are sufficient to address localizations of signals in the cerebellar cortex.

Response to comments by Reviewer #2

The authors have submitted a revised manuscript of their work showing that microglia ablation alters climbing fiber to Purkinje cell innervation in mice. The authors have provided a point by point explanation for how they addressed each comment provided by the reviewers. They have done an excellent job in addressing the comments and incorporating the changes into the paper. I have no further concerns.

We appreciate the positive evaluation of our study provided by Reviewer #2.

Response to comments by Reviewer #3

The authors have satisfactorily addressed all the comments by this reviewer

and also most of the concerns by raised by the other reviewers. They have improved their manuscript significantly.

We appreciate the positive evaluation of our study provided by Reviewer #3.